# Avoiding dynastic, assortative mating, and population stratification biases in Mendelian randomization through within-family analyses

Ben Brumpton 🔵 et al.[#]

Estimates from Mendelian randomization studies of unrelated individuals can be biased due to uncontrolled confounding from familial effects. Here we describe methods for within-family Mendelian randomization analyses and use simulation studies to show that family-based analyses can reduce such biases. We illustrate empirically how familial effects can affect estimates using data from 61,008 siblings from the Nord-Trøndelag Health Study and UK Biobank and replicated our findings using 222,368 siblings from 23andMe. Both Mendelian randomization estimates using unrelated individuals and within family methods reproduced established effects of lower BMI reducing risk of diabetes and high blood pressure. However, while Mendelian randomization estimates from samples of unrelated individuals suggested that taller height and lower BMI increase educational attainment, these effects were strongly attenuated in within-family Mendelian randomization analyses. Our findings indicate the necessity of controlling for population structure and familial effects in Mendelian randomization studies.

[#]A list of authors and their affiliations appears at the end of the paper.

Mendelian randomization is an approach that uses genetic variants as instrumental variables to estimate the causal effects of one trait (the 'exposure') on another (the 'outcome')[1–5]. It has gained popularity due to the recent expansion in the scale of genome-wide association studies (GWAS) and because it can ameliorate bias due to processes of residual confounding and reverse causation that affect most other observational approaches. In order for Mendelian randomization estimates to be valid, the genetic instrument must meet three assumptions: (1) relevance, it must associate with the exposure, (2) independence, there must be nothing that causes both the instrument and the outcome and (3) exclusion, the association of the instrument and the outcome must be entirely mediated via the exposure. Attention has been focused on developing methods to overcome bias in Mendelian randomization studies due to horizontal pleiotropy[6–11], which would violate the exclusion assumption. However, in this paper we focus on the second assumption: independence. We demonstrate how population and familial effects can violate the second assumption, and that traditional family-based methods are well placed to rectify this problem.

Mendel's laws of genetic inheritance provide a rationale for why much genetic variation for a given trait will be independent of the environment and genetic variation for other traits[1,12]. However, environmental and social factors such as assortative mating, dynastic effects, and population structure may affect the distribution of genetic variants for specific traits within populations (see Supplementary Note 1)[13–16]. Figure 1 illustrates the impact of these processes in the context of Mendelian randomization. The commonality amongst all three processes is that they induce a spurious association between the instrumenting variant and the outcome through confounding. Assortative mating occurs when partners are selected on the basis of phenotype[6,17]. For example, couples tend to have more similar education and body mass index than would be expected by chance[18,19]. If assortative mating arises due to individuals with a particular genetic predisposition selecting mates who have a particular genetically influenced phenotype, this can induce spurious genetic associations which can result in biased estimates from Mendelian randomization studies[6]. In addition, social homogamy may lead to people selecting partners who are similar to themselves[20], and this can compound across generations[6]. Dynastic effects can occur when the expression of parental genotype in the parental phenotype directly affects the offspring phenotype. For example, higher educated parents might support their children's education by providing a stimulating environment, being able to afford tutoring for their child, buying homes in better school districts, or paying for private schools. Other relationships including siblings, grandparents, uncles/aunts and cousins which may affect the offspring's phenotype can also be thought of as a likely generally weaker form of dynastic effects. Finally, residual population structure occurs when there are geographic or regional differences in allele frequency relating to a trait of interest that cannot necessarily be controlled for via principal components[13]. Confounding by population stratification[1], in which ancestry is correlated with both phenotypes and genotypes, was a major concern during the early development of Mendelian randomization[1]. However, this fear was gradually assuaged by a decade of GWAS results that were apparently reliable in the face of population structure[21]. GWAS are now performed on a huge scale; as a consequence the problem of population stratification is again of potential concern because the high statistical power of large studies renders them susceptible to bias from very subtle population structure[13,22].

Confounding in genetic association estimates, as induced by population stratification, dynastic effects and assortative mating,

can and has been resolved by using family-based study designs[6,23,24]. For example, in sibling pair studies, genetic associations at loci can be partitioned into between pair and within pair components[23]. Because genetic differences within sibling pairs reflect random independent meiotic events, within pair effects are unrelated to population stratification and most potential confounders that might influence the phenotype. Similarly, other family-based designs and within-family tests to adjust for or exploit parental genotypes exist, such as estimating maternal and offspring genetic effects using structural equation modelling[25], quantitative transmission/disequilibrium tests[26,27], or mother-father-offspring trios to adjust for parental genotypes[28]. Such within-family designs have been used to validate results from GWAS[29,30], obtain unbiased heritability estimates[31], and assess causation in the classical twin design[32,33]. Yet, despite the initial extended proposal of Mendelian randomization advising that the only way to ensure true randomization was through a within-family design[1], to-date contemporary implementations using modern genomic methods have rarely been performed. The principal reason for this has been a lack of genomic data collected from families at a scale sufficient to be suitably powered. As we now enter the age of national scale biobanks and very large twin studies, this essential extension of Mendelian randomization is becoming feasible.

This paper presents theory and simulations that demonstrate how within-family designs can be coupled with genomic data to perform Mendelian randomization analyses unbiased by population structure and family effects. We integrate these approaches in a modular fashion alongside other methods that have been developed for pleiotropy-robust inference (i.e. to be resilient to violations of the third assumption of Mendelian randomization)[7–9,34]. Using 28,777 siblings from HUNT, 32,231 siblings from the UK Biobank, and 222,368 siblings from 23andme we illustrate these methods empirically. First, we estimate the causal effect of BMI on high blood pressure and risk of diabetes as positive controls. Second, we estimate the casual effect of height and BMI on educational attainment and find substantial differences between estimates from unrelated individuals and estimates using within-family-based approaches, demonstrating the importance of controlling for family effects and population structure in Mendelian randomization studies.

## Results

**Bias evaluated using directed acyclic graphs.** There are three mechanisms depicted in Fig. 1 that induce bias in the SNP-outcome relationship of a Mendelian randomization design. The problem of population structure is well known and has been examined in detail in Lawson et al. 2019[35]. A similar confounding structure can be induced through dynastic effects because there is a path between the offspring instrument value and the offspring outcome value, which arises through parental inheritance.

Similarly, cross trait assortative mating can induce bias due to a form of collider bias, where conditioning on the assortment of parents induces a correlation between the SNP effects on $x$ and all other genetic effects on $y$. As this has been demonstrated in simulations before[6], below we demonstrate the utility of within-family designs for protecting MR estimates from bias due to familial effects, and then illustrate their importance using empirical examples. We report the analytic bias terms for the dynastic effects and assortative mating in Supplementary Notes 2 and 3.

**Simulations demonstrates robustness of the within-family design.** We conducted forward-in-time simulations to investigate bias and power related to estimating the causal effect of an

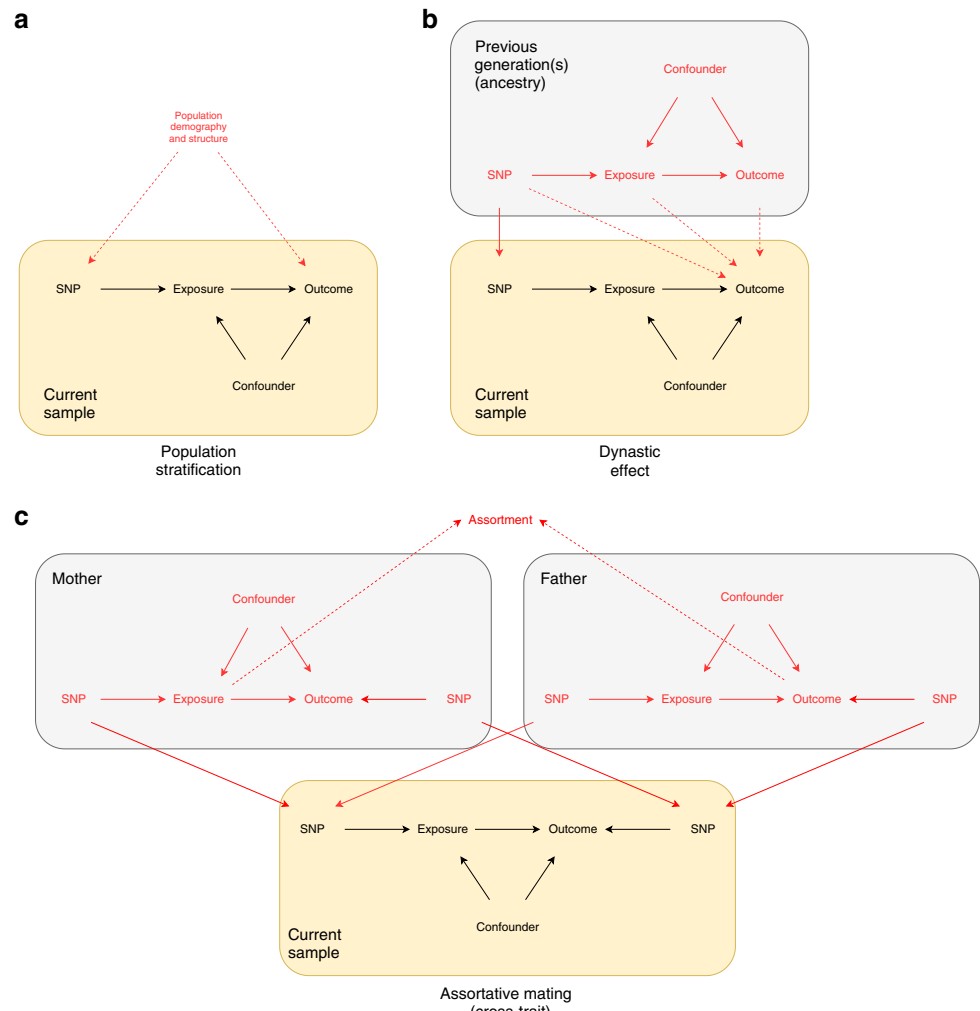

**Fig. 1 Directed acyclic graphs illustrating how population structure and familial effects can cause confound MR studies.** Black arrows indicate causal paths in the index individual, red arrows indicate causal paths in the parents, and dashed red arrows indicate confounding paths. The MR estimate of the causal effect of the exposure on the outcome is biased because of potentially unobserved confounders between the SNPs and the exposure and the outcome. **a** Illustrates how population demography and structure can confound the SNP-outcome association. **b** Illustrates how dynastic effects can induce the same statistical confounding structure of the SNP-outcome association through an entirely different mechanism. The solid red vertical arrow indicates the genetic inheritance of germline DNA. The dotted line indicates the direct (dynastic) effect of the parents on the offspring's outcomes. These can either be mediated via the exposure, the outcome or some other mechanism indicated by the direct arrow from SNP to offspring outcome. MR estimates of the effect of the exposure on the outcome in samples of unrelated individuals will be biased because there is a path between offspring SNP and the outcome via the effect of the parents' phenotypes on their offspring's outcomes (dynastic effects). The presence of dynastic effects would violate one of three key MR (instrumental variable) assumptions—the independence assumption. Estimates that control for mother or father genotype, or sibling genotype will close this path and be unbiased. **c** Illustrates how assortative mating is a third mechanism that can confound the SNP-outcome association. In this example we present cross-trait assortative mating where there is a pathway between the mother's genotype and offspring's outcome via the father's genotype for the outcome. All these forms of SNP-trait confounding can be accounted for by using methods based on within-family contrasts.

exposure on an outcome. In the simulations each parent transmits genotypes to their offspring, and the parents' exposure had a direct causal effect on the offspring's outcome (Fig. 1b). In null simulations where the exposure effect on the outcome was zero, the Mendelian randomization estimates using unrelated individuals were biased and had high false discovery rates in the presence of dynastic effects (false discovery rate > 0.75 when the confounders were $C_x$ and $C_y = 0.1$, $b_{ux} = 0.1$, $n > 10,000$). The pattern of bias in the sibling and trio methods was substantially improved, with a small amount of weak instrument bias observed, which attenuated as sample sizes improved (Fig. 2).

Where we simulated the exposure to have a causal effect on the outcome Mendelian randomization using unrelated individuals had the highest power (Fig. 2). However, the sibling

and trio design also performed well with larger sample and effect size (power > 0.9 when sample sizes ≥ 10,000, dynastic effect ≤ 0.2, effect size = 0.05). The within-family models were substantially less powerful than Mendelian randomization using unrelated individuals; as usual, controlling bias comes at a cost.

**Empirical study using HUNT, UK Biobank and 23andMe.** We begin our empirical study by using two positive control MR analyses to estimate the causal effects of two well established causal effects: BMI on diabetes and BMI on high blood pressure. We ran these analyses with and without allowing for a within family effect.

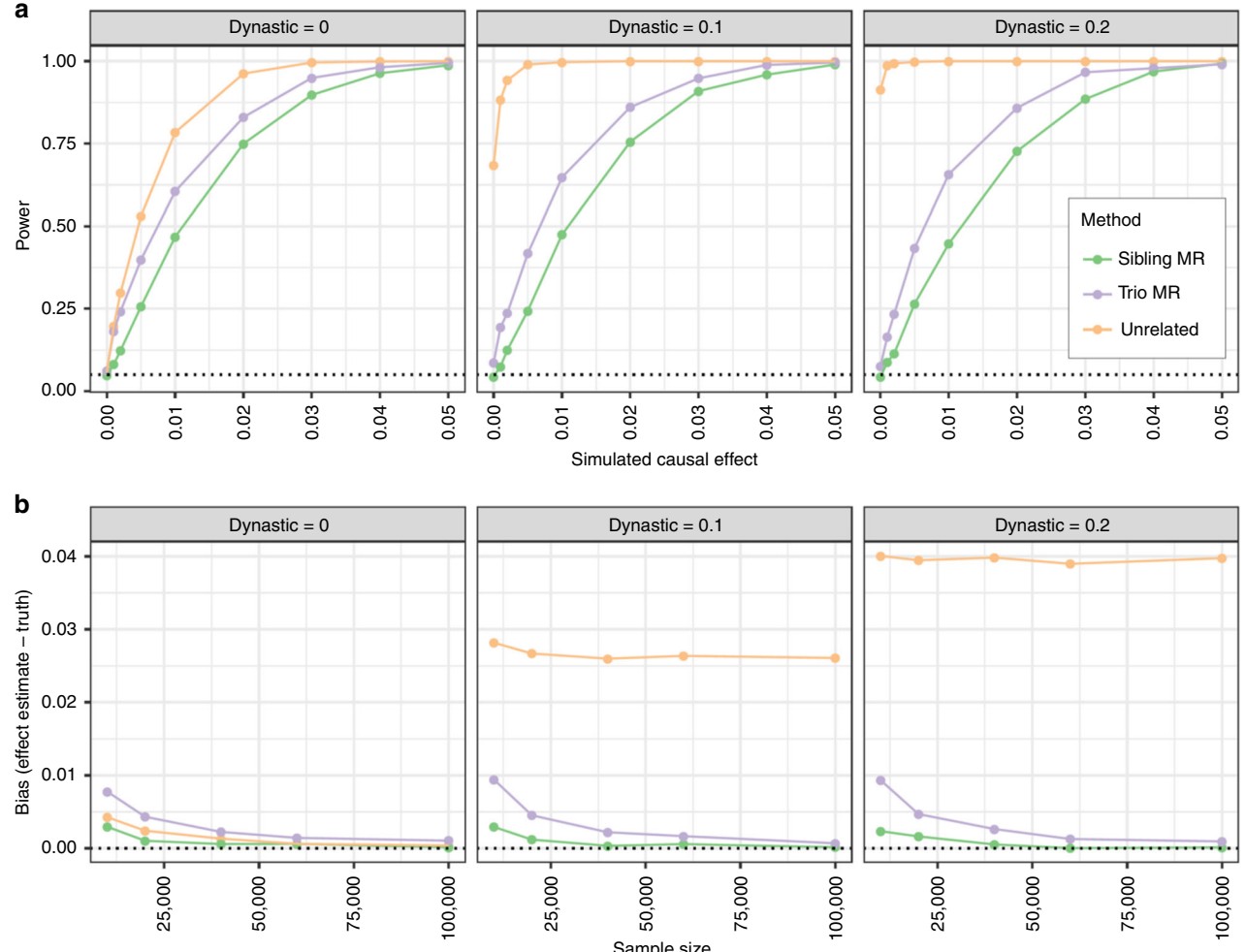

**Fig. 2 Results of simulations comparing different Mendelian randomization study designs for power and bias. a** SNP-exposure $r^2 = 0.05$; sample size = 10000 singletons, sibs, or trios; simulation involves an influence of parental exposure influencing child's confounder, which explains 10% of variance in child exposures and outcomes. For a simulated causal effect = 0, we expect the false discovery rate to be 0.05. **b** Estimated bias by sample size using different Mendelian randomization designs. The simulations are similar to **a** but allow sample size to vary and fixing the causal effect of an exposure x on an outcome y to 1% of variance explained. The bias in within-family Mendelian randomization estimates is slightly elevated when sample sizes are small due to weak instrument bias, but are otherwise are protected from the large bias seen when using unrelated samples.

Participants with higher BMI were more likely to have diabetes: each $1\,kg/m^2$ higher BMI was associated with a 0.60 (95%CI: 0.55–0.65, p-value $<1.2 \times 10^{-136}$) percentage point increase in the diabetes risk. These differences were modestly attenuated after including a family fixed effect (0.46, 95%CI: 0.40–0.52, p-value = $8.5 \times 10^{-52}$). The Mendelian randomization estimate using unrelated individuals suggested that each unit increase in BMI increased the risk of having diabetes by 0.82 (95%CI: 0.71–0.93, p-value = $3.3 \times 10^{-50}$) percentage points. This estimate remained after allowing for the fixed effects of family (1.01 percentage point increase per $1\,kg/m^2$ increase in BMI, 95%CI: 0.58–1.44, p-value = $3.3 \times 10^{-06}$). The summary data Mendelian randomization analysis allowing for family effects estimates were similar (0.75 percentage point increase per $1\,kg/m^2$ increase in BMI, 95%CI: 0.38–1.13, p-value = $7.6 \times 10^{-05}$, $p_{\text{diff unrelated}} = 0.74$). On average the associations of the SNPs with BMI and diabetes were similar before and after allowing for a family fixed effect, falling 7% (95% CI: −5% to 20%, p-value = 0.26) and increasing 11% (95%CI: −17% to 40%, p-value = 0.42) respectively.

Participants with higher BMI were more likely to have high blood pressure; each $1\,kg/m^2$ higher BMI was associated with a 2.63 (95%CI: 2.54–2.72, p-value $<1 \times 10^{-300}$) percentage point

increase in high blood pressure risk. This association did not attenuate after including a family fixed effect (2.42, 95%CI: 2.30–2.54, p-value $< 1 \times 10^{-300}$). The Mendelian randomization estimate using the sample of unrelated individuals suggested that each unit increase in BMI increased the risk of having high blood pressure by 1.59 (95%CI: 1.34–1.83, p-value = $1.3 \times 10^{-36}$) percentage points. The Mendelian randomization estimate was similar after allowing for a family fixed effect (1.13 percentage point increase per $1\,kg/m^2$ increase in BMI, 95%CI: 0.04–2.21, p-value = 0.04). The summary data Mendelian randomization estimates were similar (0.76 percentage point increase per $1\,kg/m^2$ increase in BMI, 95%CI: −0.19 to 1.70, p-value = 0.12, $p_{\text{diff unrelated}} = 0.10$). On average the associations of the SNPs and high blood pressure fell by 51% (95%CI: 23–80%, p-value = 0.0006) after allowing for family fixed effects. Overall, the associations found from the within-family studies for these positive controls establish the utility of the study design with the current scale of data.

Next, in order to examine the contrast between MR methods using unrelated individuals and family-based designs, we investigate two associations that are more liable to bias due to population structure and familial effects: height on years of

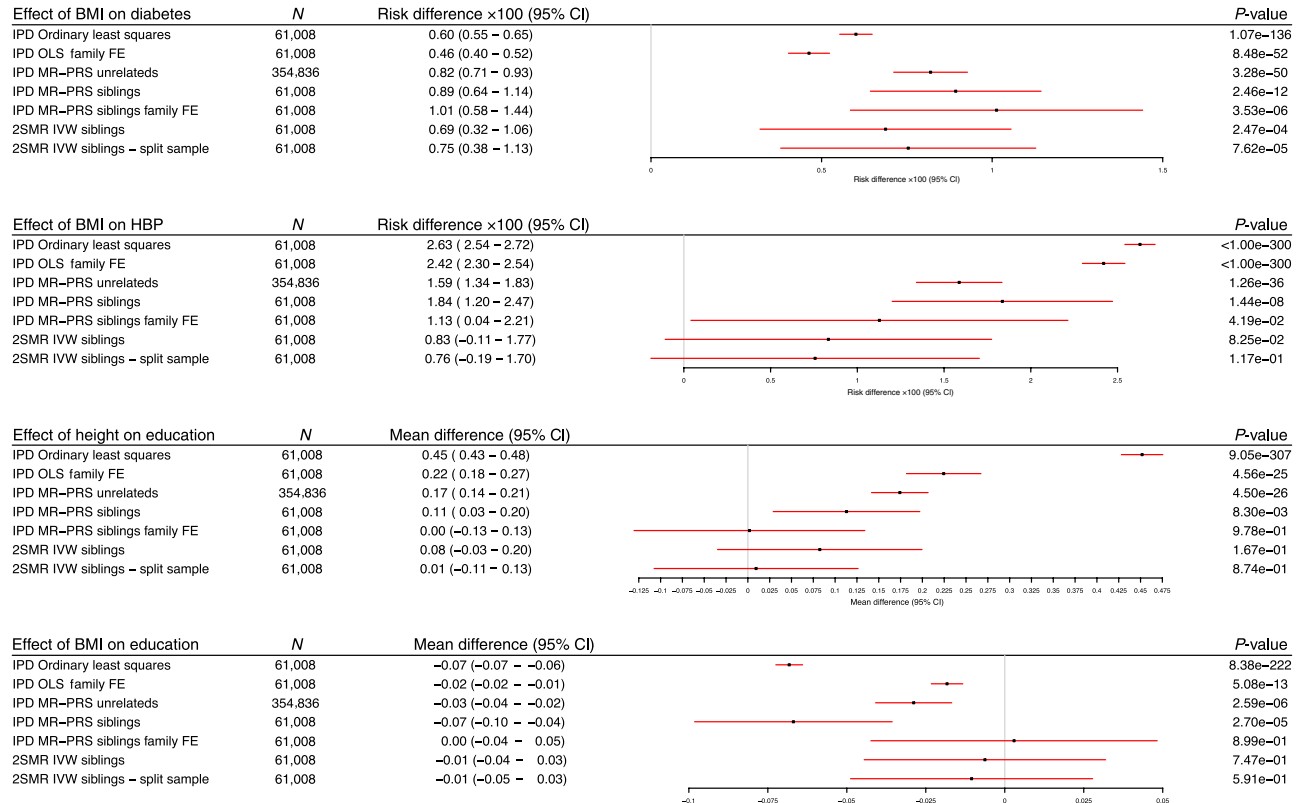

**Fig. 3 Estimates of the effect of BMI on self-reported diabetes and high blood pressure and height and BMI on educational attainment using ordinary least squares, Mendelian randomization in unrelated individuals and samples of siblings, point estimates and 95% confidence intervals reported.** All methods were consistent with higher BMI increasing diabetes and high blood pressure risk. Being taller and having lower BMI were observationally associated with higher educational attainment. The effects of height and BMI on educational attainment were attenuated but still apparent when using Mendelian randomization estimates based on unrelated individuals from HUNT and UK Biobank. The effects were eliminated after allowing for a family effect using individual-level or summary data Mendelian randomization.

education, and BMI on years of education. Taller participants were more educated; each 10 cm increase in height was associated with an additional 0.45 (95%CI: 0.43–0.48, *p*-value = $9.0 \times 10^{-307}$) years of education (Fig. 3). This association was attenuated after including a family fixed effect (0.22, 95%CI: 0.18–0.26, *p*-value = $4.6 \times 10^{-25}$). The Mendelian randomization estimate using the sample of unrelated individuals implied that each 10 cm increase in height caused an increase of 0.17 (95%CI: 0.14–0.20, *p*-value = $8.5 \times 10^{-26}$) years of education. After allowing for a family fixed effect, the Mendelian randomization estimate was greatly attenuated suggesting little evidence of a causal effect of height on education (mean difference per 10 cm increase in height: 0.002, 95%CI: −0.13 to 0.13, *p*-value = 0.98). When we used two sample Mendelian randomization by estimating the SNP-exposure and SNP-outcome associations in different samples (split sample)[36,37] and then meta-analysing, there was little evidence of a causal effect of height on education (mean difference per 10 cm increase in height = 0.009, 95%CI: −0.11 to 0.13, *p*-value = 0.87, $p_{\text{diff unrelated}} = 0.008$). On average, the associations of these SNPs with height and education fell by 18% (95%CI: 14–22%, *p*-value = $8.5 \times 10^{-24}$) and 61% (95%CI: 49–73%, *p*-value = $1.5 \times 10^{-21}$) after allowing for family fixed effects, respectively.

On average, participants with higher BMI were less educated: each 1 kg/m² higher BMI was associated with 0.07 fewer years of education (95%CI: 0.06 to 0.07, p-value = $8.4 \times 10^{-222}$, see Fig. 3). This association was attenuated after including a family fixed effect (0.02, 95%CI: 0.01 to 0.02, p-value = $5.1 \times 10^{-13}$). The Mendelian randomization estimate without allowing for

familial effects implied that each additional unit of BMI decreased years of schooling by 0.03 (95%CI: 0.02–0.04, p-value = $2.6 \times 10^{-06}$). This effect was eliminated after allowing for a family fixed effect, providing little evidence for a causal effect of BMI on educational attainment (mean difference per 1 kg/m² higher BMI = 0.00, 95%CI: −0.04 to 0.05, p-value = 0.89). Again, the effect was also largely attenuated when we used two sample summary data approaches. Using separate samples to estimate the SNP-exposure and the SNP-outcome associations allowing for family fixed effects, there was little evidence of an effect of BMI on educational attainment (mean difference per 1 kg/m² higher BMI = −0.01, 95%CI: −0.05 to 0.03, p-value = 0.59, $p_{\text{diff unrelated}} = 0.002$). On average, the association of the 69 BMI SNPs and education fell by 65% (95% CI: 34–76%, p-value = $1.8 \times 10^{-06}$) after allowing for family fixed effects. These results suggest that the methods that do not account for familial effects may be biased estimators of the individual level causal effect. We found little evidence of heterogeneity between the two sample Mendelian randomization estimates from UK biobank and HUNT, except for the effect of BMI on diabetes (p-value = 0.027).

We investigated whether our results could be explained by pleiotropy using the weighted median, weighted modal and MR-Egger estimators. These summary data Mendelian randomization estimators use estimates of the SNP-exposure and SNP-outcome associations to estimate the effect of the exposure on the outcome. These estimators are robust to a number of forms of pleiotropy. There was little evidence of differences between the inverse variance weighted (IVW) and pleiotropy robust methods,

pleiotropy from the MR-Egger intercept, or heterogeneity across the studies (Supplementary Fig. 1).

We investigated the difference (shrinkage) of the total to within family SNP-phenotype associations in HUNT and UK Biobank using seemingly unrelated regression (SUR). The estimated shrinkage is given in Supplementary Table 1. The shrinkage of the estimates suggests that accounting for familial effects affects the estimated SNP-phenotype associations for all phenotypes analysed. Educational attainment was the most strongly affected, falling by 56.8% (95%CI: 49.2–64.4%). Diabetes was the least affected falling by 11.2% (95%CI: 1.3–21.1%).

We replicated our findings using data from 223,368 individuals (111,684 families) sampled by 23andMe (Supplementary Fig. 2). There was evidence that a 1-kg m$^{-2}$ higher BMI increased the absolute individual level liability to diabetes and high blood pressure by 0.69 (95%CI: 0.37–1.00) and 1.26 (95%CI: 0.90–1.63) per 100 people, respectively. There was little evidence of heterogeneity across the weighted median, modal or MR-Egger estimators. This suggests that under a range of assumptions about pleiotropy and controlling for familial effects, BMI is likely to increase risk of diabetes and high blood pressure. In contrast, there was little evidence that height or BMI had a substantial effect on educational attainment. The results are precise and suggest that each 10 cm taller height is unlikely to increase years of education by more than 0.02 years (mean difference = 0.00, 95%CI: −0.01 to 0.02), and a 1 kg/m$^2$ unit higher BMI is unlikely to decrease years of education by more than 0.02 years (mean difference = 0.00, 95%CI: −0.02 to 0.02). The weighted median, modal and MR-Egger estimators were consistent with the IVW estimates.

## Discussion

We have presented within-family methods for Mendelian randomization and demonstrated how confounding due to population structure and familial effects can bias Mendelian randomization studies using unrelated individuals. As with most instrumental variable estimators, ceteris paribus the size of the bias induced by familial effects will be larger the smaller the individual level causal effect of the genetic variant on the exposure. The simulations illustrated how bias occurs even if the phenotype of interest has no direct causal effect on the outcome, and that these effects can theoretically induce false positive findings. The simulations further demonstrated how samples of related individuals can be used to control for these effects either using siblings or parent-offspring trios. These designs can be used in conjunction with existing approaches for accounting for horizontal pleiotropy, another potential source of bias to arise in Mendelian randomization studies. However, estimates from within-family Mendelian randomization are less precise than estimates using unrelated individuals, which is consistent with those seen for allelic association[38–40]. Compounding the issue of statistical power, there are fewer relatives than unrelated individuals in most studies. In samples from HUNT, UK Biobank and 23andMe, we investigated the impact of population structure and familial effects on four empirical examples; the effects of BMI on the risk of diabetes and high blood pressure and the effects of height and BMI on educational attainment. We found that the effects of BMI on the risk of diabetes and high blood pressure were less precise, but consistent when allowing for family effects. Conversely, the effects of height and BMI on educational attainment were almost entirely attenuated after allowing for family fixed effects, suggesting that results from previous Mendelian randomization studies using unrelated individuals may have been biased.

A substantial literature has used Mendelian randomization and samples of unrelated individuals to establish that BMI increases the risk of diabetes and high blood pressure later in life[41]. Our results suggest that confounding due to familial effects is unlikely to explain these results, and that they are more likely due to an individual level causal effect of BMI on an individual's risk. Behavioural geneticists have used longitudinal data from samples of twins to understand how different family members affect each other over time[42,43]. Other studies have used animal models to investigate how social genetic effects (i.e. indirect or dynastic effects) can affect health outcomes[44]. A rich literature has established that height and BMI are respectively positively and negatively associated with educational attainment and socio-economic position[45–47]. Consistent with our results, previous studies using twin data have indicated that the relationship between height and educational attainment is likely to be due to familial effects[48,49]. These findings raise questions about whether height and BMI have individual level causal effects on socio-economic outcomes later in life[50–52]. Our results indicate that familial effects can have important phenotypic consequences on widely studied relationships such as between height and BMI and education.

In general, within-family Mendelian randomization estimates are less precise than estimates from samples of unrelated individuals. Thus, within-family estimates of a specific association can be considered more robust, but less efficient estimates. Therefore, if there is evidence of differences between the estimates, then generally the more imprecise but less biased within-family estimates should be preferred. Our estimates of the effect of height and BMI on educational attainment are an example of this situation. If there is little evidence of differences between estimates using unrelated individuals and those allowing for family effects, then the former estimates should be preferred. Our estimates of the effect of BMI on risk of diabetes and high blood pressure are an example of this situation. This is analogous to comparing instrumental variable estimates to multivariable adjusted estimates[3]. While allowing for family fixed effects or using difference estimators will account for dynastic effects or assortative mating, these methods will not address bias due to violations of the third Mendelian randomization assumption (exclusion restriction). This assumption is that the SNPs have no direct effect of the SNPs on the outcome (i.e. no pleiotropy). MR-Egger, weighted median and mode, or Lasso estimators are robust to various forms of violations of this assumption[7–9,34]. It is trivial to use these estimators with the summary data methods we describe above and illustrate in Supplementary Figure 1. However, typically these estimators have lower power than the IVW estimator. The within-family summary data SNP-exposure and SNP-outcome associations, which allow for a family fixed effect, can be used with existing summary data estimators. Other proposed approaches allow for sophisticated control and estimation of pleiotropy and can trivially include family fixed effects[53], but again generally have lower power and require more data than Mendelian randomization approaches using allele scores or IVW. Supplementary Table 2 contains the ratio of standard errors for the MR analyses using unrelated individuals and allowing for a familial effect for the empirical results for the MR using individual participant data (IPD) and polygenic scores (MR-PRS) and two sample Mendelian randomization (2SMR) using an IVW. These estimates use identical samples, but the within family estimates had standard errors that were between 23% and 71% larger. This implies that the within family analyses would require total samples sizes between 150% and 294% larger in order to match the power of current sample sizes of unrelated individuals. A further issue concerns residual population stratification in ancestrally heterogenous GWAS such as GIANT, which may bias

SNP-phenotype associations for height, and affect analyses in other samples using SNPs identified in those GWAS[22]. Within-family Mendelian randomization can control for residual population stratification.

Within-family estimates from samples of siblings that allow for a family fixed effect are robust to biases due to dynastic effects, assortative mating and fine population structure[1,54,55]. Unlike analyses using summary data from unrelated individuals, two sample within family designs do not require the familial effects to be the same in the two samples. This is because the (different) familial effects in each sample are controlled for and the MR estimates use the individual level causal effect. Of family-based approaches, the sibling design is potentially most useful because large amounts of such data are available through biobanks and family-based studies. A limitation of current sibling designs is that they assume no sibling-sibling interaction effects. Phenotypic similarity of siblings may reflect 'passive' sharing of environments or genes, or 'active' imitation or contrast effects arising from interaction between siblings[56]. Contrast effects, which may inflate the estimated contribution of the nonshared environment in twin studies[57], can be mimicked by parental rating bias[58,59]. However, for biological phenotypes where rating bias is not a concern, Mendelian randomization could be used to study the influence via imitation or contrast of one sibling's genotype on the other's phenotype, sometimes called 'social genetic effects'[44], thereby adding to work on dynamic interplay between siblings[42,43].

Population structure, dynastic effects and assortative mating may cause bias in GWAS[14]. If a GWAS is aiming to estimate the causal effect of variants on a given phenotype, then samples of unrelated individuals may produce biased estimates and potentially spurious findings. Population structure and dynastic effects can cause bias under the null hypothesis of no effect i.e. induce spurious false positive signals. Single trait assortative mating will, however, be an unbiased test of the null hypothesis that the SNP does not affect the phenotype but will inflate SNP-phenotype associations. However, cross trait assortative mating can cause bias under the null hypothesis. Future studies could re-run GWAS on a full range of traits on samples of siblings allowing for family fixed effects. This approach would also address concerns about residual population stratification in GWAS, which may bias SNP-phenotype associations in GWAS including populations with heterogeneous ancestry[22]. However, to detect genetic variants that explain 0.1% of the variance of either the offspring or maternal effects (i.e. a 2 df test) will require sample sizes of 50,000 mother-offspring pairs to detect GWAS ($\alpha = 5 \times 10^{-08}$). Sample sizes of around 10,000 will be required to partition known loci of similar size to the above into maternal and/or offspring genetic effects ($\alpha = 0.05$)[60]. This sample size would provide valuable information about which phenotypes are likely to be most strongly affected by dynastic effects and assortative mating. It is likely that many, particularly biological, traits are relatively unaffected by these effects and thus GWAS results for these traits are unlikely to be biased due to these factors, how this requires investigation. Recent GWAS of social traits such as education reported the attenuation after allowing for family effects in their estimates in small samples[30]. Further work in this area should include estimating the consequences of familial effects for GWAS and Mendelian randomization estimates.

Population structure and familial effects can cause bias in Mendelian randomization studies. We found differences between estimates from unrelated individuals and within-family estimates in simulations and empirical analysis. The causal estimates of the effect of height and BMI on educational attainment were almost entirely attenuated after allowing for family fixed effects. Within-family methods, either using individual-level, or summary data Mendelian randomization approaches can be used to obtain unbiased estimates of the causal effects of phenotypes in the presence of dynastic effects, assortative mating and population stratification.

## Methods

**Statistical models.** We describe four methods of using family data for Mendelian randomization below. If there are only two siblings, the difference and family fixed effects methods are equivalent, see appendix for proof.

The model to be estimated can be described as:

$$x_{k,i} = \gamma_0 + \gamma_1 g_{k,i} + \gamma_2 C_{k,i} + f_k + v_{k,i} \tag{1}$$

$$y_{k,i} = \beta_0 + \beta_1 x_{k,i} + \beta_2 C_{k,i} + f_k + u_{k,i} \tag{2}$$

where $y_{k,i}$ and $x_{k,i}$ are the outcome and exposure for individual $i$ from family $k$. $g_{k,i}$ is a set of genetic variants that are associated with the exposure. $f_k$ is a family-level confounder, modelled in the empirical analysis via a family fixed effect (i.e. an indicator variable for each family). This accounts for all time invariant family-level confounders of the genetic variant-outcome association. Both $g_{k,i}$ and $f_k$ are functions of a family-genetic component. $u_{k,i}$ and $v_{k,i}$ are random error terms. $C_{k,i}$ is a confounder of the association of the exposure and the outcome, $\gamma_2$ and $\beta_2$ indicate the effect of the confounder on the exposure and the outcome. $\beta_1$ is the true causal effect of the exposure on the outcome which we wish to estimate. This model implies that Mendelian randomization using data from unrelated individuals would produce biased estimates of $\beta_1$ due to the correlation between $g_{k,i,j}$ and $f_k$. The effect of the exposure on the outcome can be estimated using individual level data allowing for a family fixed effect, or summary level data using difference methods within families, or by allowing for a family fixed effect. We describe these approaches below.

**Siblings difference method.** To apply Mendelian randomization to samples of siblings, effect estimates for the SNP-exposure association and SNP-outcome association are based on correlating the phenotypic divergence with the genotypic divergence within sibling pairs. Taking the difference between siblings removes the effect of the family-level confounder. For any pair of siblings within family $k$, indicated $k,1$ and $k,2$, the genotypic difference at genetic variant $j$ is:

$$\delta_{k,j} = g_{k,1,j} - g_{k,2,j} \tag{3}$$

The association between the genotypic differences and phenotypic differences in the exposure, $x$, and outcome $y$, for SNP $j$ can be estimated via:

$$x_{k,1} - x_{k,2} = \gamma_j \delta_{k,j} + \dot{v}_{k,j} \tag{4}$$

$$y_{k,1} - y_{k,2} = \Gamma_j \delta_{k,j} + \dot{u}_{k,j} \tag{5}$$

The estimated associations, $\gamma_j$ and $\Gamma_j$, can be used with any summary level Mendelian randomization estimator. Here we apply the inverse variance weighted (IVW) approach. Each pair of siblings can be included as a separate pseudo-independent pair.

**Family fixed effect with sibling data.** Alternatively, we can estimate the associations using family fixed effects indicated by $f_k$ for each family, which is equivalent to centring the data by subtracting the family mean.

$$x_{k,i} = \gamma_0 + \gamma_{1,j} g_{k,i,j} + f_k + \ddot{v}_{k,i,j} \tag{6}$$

and

$$y_{k,i} = \beta_0 + \Gamma_1 g_{k,i,j} + f_k + \ddot{u}_{k,i,j}. \tag{7}$$

This estimator accounts for any differences between families, which includes any effect of assortative mating or dynastic effects common to all siblings by including a dummy variable for each family. This provides unbiased estimates of the SNP-exposure and SNP-outcome associations. These estimates can be used with standard summary data Mendelian randomization methods. The difference and family fixed effects methods are identical if there are only two siblings in each family. This fact follows from substituting equations iv and v into equations ii and iii and simplifying (see "Appendix" for proof). If there are more siblings, then the estimators are non-identical, but likely to be similar, see the appendix for further details. An analytically convenient method to use for this estimator is the within transformation. The within transformation either de-means the variables for or additionally adjusts for the family level means. Demeaned using the within transformation is computationally efficient, particularly for large sample sizes—and is the analytic method used by many statistical packages for fixed effects estimators. An advantage of further adjusting for the within family mean is that it provides an estimate of the between family effect. Cluster robust standard errors can be used to allow for clustering and relatedness within families.

**Adjusting for parental genotype with mother-father-offspring trio data.** Finally, if data on mother-father-offspring trios are available, the estimates of the SNP-exposure and SNP-outcome associations for each child can be adjusted for

their mother's and father's genotypes, indicated by $g_{im,j}$ and $g_{if,j}$ respectively[61]:

$$x_i = \gamma_0 + \gamma_{1,j}g_{i,j} + \gamma_{2,j}g_{im,j} + \gamma_{3,j}g_{if,j} + u_{i,j} \quad (8)$$

and

$$y_i = \beta_0 + \Gamma_1 g_{i,j} + \Gamma_2 g_{im,j} + \Gamma_3 g_{if,j} + v_{i,j} \quad (9)$$

Again, these associations can be used to estimate the effect of the exposure on the outcome using summary data Mendelian randomization methods. It is possible to estimate the effect of offspring genotype on the exposure and outcome conditional on the mother and father genotype using summary data[25,61]. The estimated causal effect can be biased if both the SNP-exposure and SNP-outcome associations are estimated in the same sample[62]. This bias can be eliminated by splitting the sample and estimating the associations in separate samples.

**Two-stage least squares with sibling data**. Many summary data methods assume no measurement error on the SNP-exposure association (NOME)[63]. This assumption may lead to underestimation of the standard error of the effect of the exposure on the outcome. Two-stage least squares can estimate the effect of the exposure on the outcome using the individual-level data from siblings. Estimators that use individual-level data can integrate the estimation error from the SNP-exposure association. We used cluster robust standard errors that allow for clustering and relatedness within family. We used the commands xtivreg and plm[64].

**Simulation of dynastic effects**. We simulated a cohort consisting of pairs of unrelated mothers and fathers who had two offspring. All individuals had a genome of 90 SNPs. We set the distribution of identity by descent (IBD) across the 90 SNPs as $N$ (0.5,0.037) for each sibling pair, as per theory, because there are on average 90 recombination events separating human siblings. Hence, we assume that each SNP has an independent effect on the exposure.

We defined parents' exposure and outcome by defining confounders $u$, exposure $x$ and outcome $y$. A directed acyclic graph illustrating these relationships is shown in Fig. 1b. The confounder influences the parents' exposure and outcome. The offspring have the same confounding structure, except the parent's exposure affects their offspring's outcomes via a dynastic or ancestry effect. The genetic influence of each of the 90 SNPs on the exposure amounts to explaining $V_{gx}$ of the variance in the exposure. We assumed no horizontal pleiotropy. All estimates assume $V_{gx} = 0.1$ and 90 independent causal variants (i.e. somewhat similar to GIANT results for BMI)[65].

To generate the phenotypes under a model of dynastic effects, the offspring outcome was influenced by both the offspring exposure and the parents' exposures. In these simulations all phenotypes had mean of 0 and variance of 1. Differing strengths of dynastic effects by which the parental exposure influenced the offspring outcome were generated ($b_{ux} = 0,0.01,0.02$) under a set of models with a range of causal effects of the exposure on the outcome ($b_{xy} = 0,0.001,0.002,0.005,0.01,0.05$). We calculated the false discovery rate (proportion of test with p-value < 0.05) for 100 iterations of each simulation using each of three methods: standard IVW as applied to one of each individual in a set of siblings (i.e. a sample of unrelated individuals), the within-family sibling design, and the within-family trio design. Finally, we calculated bias (estimated effect— simulated effect) for all three study approaches by simulated confounding ($C_x$ and $C_y = 0,0.1,0.2$), dynastic bias ($b_{Ux} = 0,0.1,0.2$) and simulated causal effects ($b_{xy} = 0,0.001,0.002,0.005,0.01,0.05$). The sample sizes were 10,000, 20,000, 40,000, 60,000 and 100,000 sibling pairs for all simulations.

If the familial effect influences the exposure, but does not affect the outcome, then we would not expect bias in the Mendelian randomization analysis. This is because there would be no open path between the SNP and the outcome. This is illustrated in the directed acyclic graph illustrated in Supplementary Fig. 3.

**Empirical analysis**. To demonstrate the approach and assess potential bias from population structure and familial effects, we conducted within-family Mendelian randomization using two illustrative examples in the HUNT and the UK Biobank[66–68]. We estimated the effects of BMI on high blood pressure and diabetes and the effects of height and BMI on educational attainment. The effects of BMI on diabetes and high blood pressure have been well studied and provide a positive control[41]. These effects on clinical outcomes experienced later in life are unlikely to be due to assortative mating or dynastic effects, because parents are less likely to assort on genetic liability for diabetes or high blood pressure. The genetic liability for these conditions was probably unknown when the couples were formed. Previous longitudinal and Mendelian randomization studies using unrelated individuals have suggested that height and BMI may affect educational attainment[45,50]. Such an association, if causal, might be counteracted by changing educational policy. However, the association may be due to parents' education, via dynastic effects or assortative mating, where more educated people select taller and slimmer partners. Assortative mating and dynastic effects can confound the association between genetic variants when data from the offspring generation are used. Therefore, the ratio of individual-level causal effects to family-level effects is likely to be higher for the effects of BMI on clinical end points than for the effects of height and BMI on education.

**The Nord-Trøndelag Health Study**. The Nord-Trøndelag Health Study (HUNT) is a population-based cohort study. The study was carried out at four time points over approximately 40 years (HUNT1 [1984-1986], HUNT2 [1995-1997] and HUNT3 [2006-2008] and HUNT4 [2017-2019]). A detailed description of HUNT is available[66]. We include 71,860 participants from HUNT2 and HUNT3 as they have been recently genotyped using one of three different Illumina HumanCoreExome arrays (HumanCoreExome12 v1.0, HumanCoreExome12 v1.1 and UM HUNT Biobank v1.0). For a flow chart of participants inclusion and exclusion from the study see Supplementary Figure 4. Imputation was performed on samples of recent European ancestry using Minimac3 (v2.0.1, http://genome.sph.umich.edu/wiki/Minimac3) from a merged reference panel constructed from the Haplotype Reference Consortium (HRC) panel (release version 1.1) and a local reference panel based on 2202 whole-genome sequenced HUNT participants[12–14]. Ancestry of all samples was inferred by projecting all genotyped samples into the space of the principal components of the Human Genome Diversity Project (HGDP) reference panel (938 unrelated individuals; downloaded from http://csg.sph.umich.edu/chaolong/LASER/)[15,16], using PLINK. We defined recent European ancestry as samples that fell into an ellipsoid spanning exclusively the European population within the HGDP panel. We restricted the analysis to individuals of recent European ancestry who passed quality control. Among these, 17,329 pairs of siblings comprising of 28,777 siblings, were inferred using KING[17], where an estimated kinship coefficient between 0.177 and 0.355, the proportion of the genomes that share two alleles IBD > 0.08, and the proportion of the genome that share zero alleles IBD > 0.04 corresponded to a full sibling pair.

**HUNT descriptive data**. There were 56,374 genotyped individuals in HUNT, including 11,448 families with at least two siblings comprising of 28,777 siblings (14,718 women) with complete data on genotype, height and education, diabetes and blood pressure (see supplementary figure 5 for a flow chart of participant inclusion and exclusion). On average the participants in the full unrelated sample were 48.7 (SD = 15.1) years old, had a BMI of 26.3 kg/m$^2$ (SD = 3.9), were 177.5 cm tall (SD = 6.6) and 164.3 cm tall (SD = 6.1) for men and women respectively, 2.5% of them had diabetes, and 42.5% had high blood pressure and had 12.3 years (SD = 2.3) of education. High blood pressure was defined as either currently taking anti-hypertensive medication or having systolic or diastolic blood pressure above 140 mmHg or 90 mmHg on average across up to three measurements in HUNT2.

**Questionnaires, clinical measurements and hospitalizations**. Participants attended a health survey which included comprehensive questionnaires, an interview and clinical examination. The participants' height and weight were measured with the participant wearing light clothes without shoes to the nearest centimetre and half kilogram, respectively. Education was defined using the question 'What is your highest level of education'. Participants answered one of five categories (1) primary school, (2) high school for 1 or 2 years, (3) complete high school, (4) college or university less than 4 years, and (5) college or university 4 years or more. Participants with university degrees were assigned to 16 years of education, those who completed high school were assigned 13 years, those who attended high school for 1 or 2 years were assigned 12 years, and those who only attended primary school were assigned 10 years. Diabetes was defined using responses to the question 'Have you had or do you have diabetes?', which has high validity[69]. High blood pressure was defined as those with systolic or diastolic blood pressure equal to or more than 140 or 90 mmHg, respectively, or reported use of antihypertensive medication.

**Ethics**. This study was approved by the Regional Committee for Medical and Health Research Ethics, Central Norway and all participants gave informed written consent (application numbers 2015/1209, 2015/2292 and 2017/2479).

**The UK Biobank**. The UK Biobank invited over 9 million people and sampled 503,317 participants from March 2006 to October 2010. The study sampled individuals from 21 study centres across Great Britain. A detailed description of the study can be read elsewhere[67,68,70]. The participants gave blood samples, from which DNA was extracted. Full details of the quality control process are available elsewhere[71]. Briefly, we excluded participants who had mismatched genetic and reported sex, or those with non-XX or XY chromosomes, extreme heterozygosity or missingness. We used variants in the Haplotype Reference Consortium (HRC) panel.

**The UK Biobank descriptive data**. There were up to 370,180 genotyped individuals in the UK Biobank, among whom were 16,847 families with at least two siblings, with 33,642 siblings (19,445 women) with complete data on genotype, height and education, diabetes and blood pressure. We restricted the analysis to siblings born in England to ensure that they experienced a similar school system. For a flow chart of participants inclusion and exclusion from the study see Supplementary Fig. 5. On average, the participants without siblings were 57.5 (SD = 7.4) years old, had a BMI of 27.4 kg m$^{-2}$ (SD = 4.8), were 175.0 cm tall (SD = 6.8) and 162.8 cm tall (SD = 6.2) for men and women respectively, had 14.1 (SD = 2.3) years of education. 4.5% of them had diabetes and 54.0% had high blood pressure. High blood pressure was defined as either having a diagnosis of high blood

pressure or having systolic or diastolic blood pressure above 140 mmHg or 90 mmHg respectively on average across up to two clinic measurements.

**Questionnaires, clinical measurements and hospitalizations**. Weight (ID:21002) and standing height (ID:50) were measured using standardised instruments the baseline assessment centre visits. We defined education using the participants' response to the touch screen questionnaires about their educational qualifications (ID = 6138). We defined educational attainment using the participants' highest reported educational qualification at either measurement occasion. We assigned participants with university degrees to 17 years of education, those with professional qualifications such as teaching or nursing to 15 years, those with A-levels to 14 years, those with National Vocational Qualifications (NVQs), Higher National Diplomas (HNDs) to 13 years, General Certificate of Secondary Education (GCSEs), Certificate of Secondary Education (CSEs) or O-levels to 12 years, and those who reported no qualifications to 11 years, which was the legal minimum length of education for this cohort. Diabetes and high blood pressure were defined using responses to the self-reported touch screen questionnaire (ID = 6150 and ID = 2443). We used self-reported measures because measured blood pressure is affected by medication use. Missing values at the baseline visit were replaced by measures from subsequent visits if available.

**Ethics**. UK Biobank received ethical approval from the Research Ethics Committee (REC reference for UK Biobank is 11/NW/0382). This research was approved as part of application 8786.

**23andMe replication**. Individuals in the 23andMe replication dataset were customers of 23andMe, Inc., a personal genomics company. The 23andMe study protocols were approved by an external AAHRPP-accredited institutional review board and conducted in accordance with the Declaration of Helsinki principles.

There were 111,684 sibling pairs in the 23andMe dataset for a total of 223,368 genotyped individuals with complete data on genotype, height, BMI, education, diabetes, and blood pressure. For a flow chart of participants inclusion and exclusion from the study see Supplementary Fig. 6. Participants self-reported their mass (in kilograms) and height (in metres), from which BMI was calculated. To determine years of education, participants were asked 'What is the highest degree or level of school you have completed? If currently enrolled, please select the previous grade or highest degree received'. The following response options were then mapped onto the corresponding years of education: less than high school = 10 years, high school diploma or equivalency (GED) = 12 years, Associate's degree (for example, AA, AS) = 14 years, Vocational degree = 14 years, some college but no degree = 14 years, Bachelor's degree (for example, BA, BS) = 16 years, Master's degree (for example, MA, MS, MEng, MEd, MSW, MBA) = 19 years, Professional degree beyond a Bachelor's degree (for example, MD, DDS, DVM, LLB, JD) = 19 years, doctoral degree (for example, PhD, EdD) = 22 years. Research participants self-reported having ever been diagnosed or treated for both Type II diabetes or high blood pressure.

As previously described[73], DNA extraction and genotyping were performed on saliva samples by National Genetics Institute. Samples were genotyped on one of five Illumina-based genotyping platforms. Samples had minimum call rates of 98.5%. We phased participant data using either an internally developed tool, Finch (V1-V4 genotyping arrays) or Eagle2 (V5 genotyping array)[74]. We imputed phased research participant data using Minimac3 and a reference panel that combined both the May 2015 release of the 1000 Genomes Phase 3 haplotypes with the UK10K imputation reference panel (n = 6285). Throughout, we treated structural variants and small indels the same as SNPs. We used the same list of SNPs as for HUNT and UK Biobank, restricted to those that passed the 23andMe genotypic data QC.

We computed association test results using the sibling difference method assuming additive allelic effects logistic regression for case-control exposures, linear regression for quantitative exposures.

**Genotype arrays**. As previously described[73], samples were genotyped on one of five genotyping platforms. The v1 and v2 platforms were variants of the Illumina HumanHap550+ BeadChip, including about 25,000 custom SNPs selected by 23andMe, with a total of about 560,000 SNPs. The v3 platform was based on the Illumina OmniExpress+ BeadChip, with custom content to improve the overlap with our v2 array, with a total of about 950,000 SNPs. The v4 platform was a fully customized array, including a lower redundancy subset of v2 and v3 SNPs with additional coverage of lower-frequency coding variation, and about 570,000 SNPs. The v5 platform is an Illumina Infinium Global Screening Array (~640,000 SNPs) supplemented with ~50,000 SNPs of custom content.

**The Finch phasing algorithm**. As previously described[73], Finch implements the Beagle haplotype graph-based phasing algorithm, modified to separate the haplotype graph construction and phasing steps[75]. It extends the Beagle model to accommodate genotyping error and recombination, to handle cases where there are no consistent paths through the haplotype graph for the individual being phased. We constructed haplotype graphs for European and non-European samples on each 23andMe genotyping platform from a

representative sample of genotyped individuals, and then performed out-of-sample phasing of all genotyped individuals against the appropriate graph. For the X chromosome, we built separate haplotype graphs for the non-pseudo autosomal region and each pseudo autosomal region, and these regions were phased separately.

**Imputation panel generation**. As previously described[73], imputation panels created by combining multiple smaller panels have been shown to give better imputation performance than the individual constituent panels alone[76]. To that end, we combined the May 2015 release of the 1000 Genomes Phase 3 haplotypes with the UK10K imputation reference panel to create a single unified imputation reference panel[77,78]. To do this, multiallelic sites with N alternate alleles were split into N separate biallelic sites. We then removed any site whose minor allele appeared in only one sample. For each chromosome, we used Minimac3 to impute the reference panels against each other, reporting the best-guess genotype at each site[79]. This gave us calls for all samples over a single unified set of variants. We then joined these together to get, for each chromosome, a single VCF with phased calls at every site for 6285 samples.

**Imputation**. In preparation for imputation we split each chromosome of the reference panel into chunks of no more than 300,000 variants, with overlaps of 10,000 variants on each side. We used a single batch of 10,000 individuals to estimate Minimac3 imputation model parameters for each chunk. We imputed phased participant data against the chunked merged reference panel using Minimac3, treating males as homozygous pseudo-diploids for the non-pseudo autosomal region.

**Selection of genotypes for instruments**. For the analysis of HUNT and UK Biobank we selected 385 independent ($r^2 < 0.01$ within 10,000 kb) SNPs associated with height ($p < 5 \times 10^{-08}$) from Wood et al. and 79 associated with BMI in Locke et al.[65,72]. Neither HUNT nor UK Biobank were included in these studies. We clumped variants using the TwoSampleMR package[80]. We harmonised the alleles' effect sizes across the two samples and constructed weighted polygenic scores which were sums of the phenotype increasing alleles and weighted each variant by its effect on the phenotype in the published GWAS. For the analysis of the 23andMe data we used the subset of 347 and 64 SNPs available in the 23andMe data.

**Empirical analyses**. We compared seven empirical estimates of the effect of BMI on self-reported diabetes and high blood pressure and the effects of height and BMI on educational attainment. We used the familial fixed effects models (2) described above:

1. IPD ordinary least squares (OLS): The multivariable adjusted phenotypic association using ordinary least squares. Estimated using reg/plm commands.
2. IPD OLS family fixed effects (FE): The multivariable adjusted phenotypic association using ordinary least squares allowing for a family fixed effect across siblings. Estimated using xtreg/plm commands.
3. IPD MR-PRS unrelateds: A standard MR estimate of the effect of each exposure on the outcomes using the largest available sample of unrelated individuals. These models do not allow for any familial effects and are likely to suffer from bias. The estimate uses a polygenic score for each exposure and two stage least squares. Estimated separately in HUNT and UK Biobank using the ivreg and ivreg2 packages.
4. IPD MR-PRS siblings: As with 3. above, but restricted to siblings. Estimated using the ivreg2 package. We include this estimate to demonstrate the impact of including a family fixed effect on the estimates when holding the sample constant.
5. IPD MR-PRS siblings family fixed effects: An estimate using individual level data from the full sample of siblings allowing for family fixed effects. This allows for familial effects. Estimated separately in UK Biobank and HUNT using the xtivreg and plm packages[64]. This equivalent to family fixed effects models (vi and vii) described above and uses the same sample as 4. above.
6. 2SMR IVW siblings: An estimate using SNP summary data for Mendelian randomization including family fixed effects. The SNP-exposure and SNP-outcome associations were estimated on the same sample. The SNP level estimates of the effect of the exposure on the outcome were estimated separately in HUNT and UK Biobank and the overall Mendelian randomization (Wald) estimates are calculated for each SNP. For each SNP the Wald estimate is the ratio of the SNP-outcome and SNP-exposure association. We combine the estimate using random effects Inverse Variance Weighted (IVW) meta-analysis. This demonstrates how within family association estimates can be used within the two-sample MR framework.
7. 2SMR IVW siblings—split sample: As with 6. above, but the SNP-exposure and SNP-outcome associations were estimated in separate samples (i.e. split sample approach). The overall Wald estimates were combined via IVW meta-analysis as above. This ensures that there is no sample overlap between

the samples used to estimate the SNP-exposure and SNP-outcome associations. This eliminates the risk of weak instrument/sample overlap bias.

**Covariates and standard errors**. All analyses included age, sex, and the first 20 principal components of genetic variation. Cluster robust standard errors were used to allow for heteroskedasticity and allow for clustering and relatedness across siblings within families. Inclusion of the covariates age, sex, and principal components did not meaningfully affect the within family estimates, as they are independent of genotype conditional on sibling genotype. However, including these covariates may absorb some of the variation in the outcome and increase the precision of our estimates.

**Sensitivity analyses**. Finally, we tested for difference ($p_{\text{diff unrelated}}$) between the Mendelian randomization estimates using the unrelated individuals and the summary data within-family estimates using the split sample approach (i.e. as in 6. above)[81]. We investigated whether our results could be explained by pleiotropy using the weighted median, weighted modal and MR-Egger estimators and the SNP-phenotype associations allowing for a family fixed effect[8,10,82]. We used a split sample approach in which the SNP-exposure and SNP-outcome associations were estimated in separate samples. We estimated the percentage change in the SNP-phenotype coefficients with and without allowing for a family fixed effect.

**Shrinkage**. We investigated shrinkage of the total to within family SNP-phenotype associations using seemingly unrelated regression. We estimated the shrinkage for each of the 455 SNPs included in the analysis and the five phenotypes (education, BMI, height, diabetes and high blood pressure). We then meta-analysed estimates for each SNP and phenotype across the two studies. Finally, independently for each phenotype we meta-analysed across all 455 SNPs used in the analysis to give an average shrinkage for all SNPs.

## Data availability

Data from the HUNT study was accessed under ethics approvals 2015/1209 REK midt 2015/2292 REK midt, and 2017/2479 REK sør-øst, and project number 2019/2181. Data from the UK Biobank was accessed as part of application 8786. The empirical datasets used with the HUNT study and UK Biobank will be archived with the studies and will be made available to individuals who obtain the necessary permissions from the studies' data access committees. Data from 23andMe was processed by 23andMe, and the individual level data cannot be made publicly available, however, 23andMe do provide access to summary data via a system of managed access. If you would like to apply for access, please see the following website for more details https://research.23andme.com/dataset-access/.

## Code availability

The code used to clean and analyse the data and the SNP level summary statistics are available here: https://github.com/nmdavies/within_family_mr. Code for the simulations are available here: https://github.com/mrcieu/mrtwin_power.

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

## Acknowledgements

Jonathan Beauchamp provided valuable comments and suggestions on an earlier draft of this paper. This research has been conducted using the UK Biobank Resource under Application Number 16729. Quality Control filtering of the UK Biobank data was conducted by R.Mitchell, G.Hemani, T.Dudding, L.Paternoster as described in the published protocol (doi:10.5523/bris.3074krb6t2frj29yh2b03x3wxj). The MRC IEU UK Biobank GWAS pipeline was developed by B.Elsworth, R.Mitchell, C.Raistrick, L. Paternoster, G.Hemani, T.Gaunt (doi: 10.5523/bris.pnoat8cxo0u52p6ynfaekeigi). The Medical Research Council (MRC) and the University of Bristol support the MRC Integrative Epidemiology Unit [MC_UU_00011/1].

N.M.D. is supported by an Economics and Social Research Council (ESRC) Future Research Leaders grant [ES/N000757/1] and a Norwegian Research Council Grant number 295989. JHB was funded by the Norwegian Research Council with grant number 295989. DME is funded by a National Health and Medical Research Council Senior Research Fellowship (1137714). E.M.T.D. was supported by NIH grants R01AG054628 and R01HD083613, and by the Jacobs Foundation. L.D.H. is supported by a Career Development Award from the UK Medical Research Council (MR/M020894/1). This work is part of a project entitled 'social and economic consequences of health: causal inference methods and longitudinal, intergenerational data', which is part of the Health Foundation's Social and Economic Value of Health Research Programme (Award 807293). The Health Foundation is an independent charity committed to bringing about better health and health care for people in the UK. G.A.V. is supported by a Norwegian Research Council grant code 250335. C.A.R. receives support from the National Institutes of Health (NIH) including R01AG060470, R01AG059329, R01AG058068, R01AG018386, and R01AG046938. NLP receives funding from the National Institutes of

Health Grants No. R01AG060470, R01AG059329. The Nord-Trøndelag Health Study (The HUNT Study) is a collaboration between HUNT Research Center (Faculty of Medicine and Health Sciences, NTNU, Norwegian University of Science and Technology), Nord-Trøndelag County Council, Central Norway Regional Health Authority, and the Norwegian Institute of Public Health. The K.G. Jebsen Center for Genetic Epidemiology is funded by Stiftelsen Kristian Gerhard Jebsen; Faculty of Medicine and Health Sciences, NTNU; The Liaison Committee for education, research and innovation in Central Norway; and the Joint Research Committee between St. Olavs Hospital and the Faculty of Medicine and Health Sciences, NTNU. The genotyping in HUNT was financed by the National Institute of Health (NIH); University of Michigan; The Research Council of Norway; The Liaison Committee for education, research and innovation in Central Norway; and the Joint Research Committee between St. Olavs Hospital and the Faculty of Medicine and Health Sciences, NTNU. J.K. has been supported by the Academy of Finland (grants 308248, 312073). R.M.F. and R.N.B. are supported by Sir Henry Dale Fellowship (Wellcome Trust and Royal Society grant: WT104150). G.H. is supported by the Wellcome Trust and Royal Society [208806/Z/17/Z]. A.H. was funded by the South-Eastern Norway Regional Health Authority, grants 2018059 and 2020022. We thank the customers of 23andMe who answered surveys and participated in this research. No funding body has influenced data collection, analysis or its interpretation. This publication is the work of the authors, who serve as the guarantors for the contents of this paper.

## Author contributions

G.D.S., N.M.D., K.H. and B.M.B. obtained funding for this study. B.M.B., G.H. and N.M.D. designed, analysed and cleaned the data, interpreted results, wrote and revised the paper. G.H. ran the simulation study. E.S. provided the proof comparing the difference and fixed effects estimators, interpreted results and revised the paper. K.H. ran the replication in 23andMe data, drafted the sections relating to the 23andMe data and revised the paper. G.D.S. conceived of the study, interpreted the results, wrote and revised the paper. F.P.H., S.H., G.A.V., Y.C., L.D.H., A.H., D.I.B., A.H., J. H., M.N., M.G.N., N.L.P., C.A.R., E.M.T.-D., A.G., L.H., T.M., S.L., A.A., F.W., W-M. C., J.H.B., K.H., C.W., D.M.E., J.K. and B.O.A. interpreted the results and revised the paper.

## Competing interests

K.H., A.A. and members of the 23andMe Research Team are employees of and have stock, stock options, or both, in 23andMe. The other authors declare no competing interests.

## Additional information

Ben Brumpton [1,2,3✉], Eleanor Sanderson [2,4], Karl Heilbron[5], Fernando Pires Hartwig [2,6], Sean Harrison [2,4], Gunnhild Åberge Vie [1], Yoonsu Cho [2,4], Laura D. Howe[2,4], Amanda Hughes[2,4], Dorret I. Boomsma[7], Alexandra Havdahl [2,8,9], John Hopper[10], Michael Neale [11], Michel G. Nivard [7], Nancy L. Pedersen[12], Chandra A. Reynolds[13], Elliot M. Tucker-Drob[14], Andrew Grotzinger [14], Laurence Howe[2,4], Tim Morris [2,4], Shuai Li[10,15], The Within-family Consortium*, The 23andMe Research Team*, Adam Auton[5], Frank Windmeijer[2,16], Wei-Min Chen[17], Johan Håkon Bjørngaard[1,18], Kristian Hveem[1], Cristen Willer [19,20,21], David M. Evans [2,22], Jaakko Kaprio [23,24], George Davey Smith [2,4,26], Bjørn Olav Åsvold[1,25,26], Gibran Hemani [2,4,26] & Neil M. Davies [1,2,4,26✉]

[1]K.G. Jebsen Center for Genetic Epidemiology, Department of Public Health and Nursing, NTNU, Norwegian University of Science and Technology, Trondheim, Norway. [2]Medical Research Council Integrative Epidemiology Unit, University of Bristol, BS8 2BN Bristol, UK. [3]Clinic of Thoracic and Occupational Medicine, St. Olavs Hospital, Trondheim University Hospital, Trondheim, Norway. [4]Population Health Sciences, Bristol Medical School, University of Bristol, Barley House, Oakfield Grove, Bristol BS8 2BN, UK. [5]23andMe, Inc., 223 N Mathilda Avenue, Sunnyvale, CA 94086, USA. [6]Postgraduate Program in Epidemiology, Federal University of Pelotas, Pelotas, Brazil. [7]Netherlands Twin Register, Department of Biological Psychology, Vrije Universiteit Amsterdam, Amsterdam, The Netherlands. [8]Nic Waals Institute, Lovisenberg Diaconal Hospital, Spångbergveien 25, 0853 Oslo, Norway. [9]Department of Mental Disorders, Norwegian Institute of Public Health, Sandakerveien 24 C, 0473 Oslo, Norway. [10]Centre for Epidemiology and Biostatistics, Melbourne School of Population and Global Health, The University of Melbourne, 207 Bouverie Street, Carlton, VIC 3053, Australia. [11]Virginia Institute for Psychiatric and Behavior Genetics, Virginia Commonwealth University, Richmond, VA, USA. [12]Department of Medical Epidemiology and Biostatistics, Karolinska Institutet, Stockholm, Sweden. [13]Department of Psychology, University of California Riverside, Riverside, CA, USA. [14]Department of Psychology and Population Research Center, University of Texas at Austin, 108 E. Dean Keeton Stop A8000,, Austin, TX 78712, USA. [15]Centre for Cancer Genetic Epidemiology, Department of Public Health and Primary Care, University of Cambridge, Strangeways Research Laboratory, Worts Causeway, Cambridge CB1 8RN, UK. [16]Department of Economics, University of Bristol, 2 Priory Road, Bristol BS8 1TU, UK. [17]Center for public health genomics, Department of public health sciences, University of Virginia, Charlottesville, VA, USA. [18]Faculty of Nursing and Health Sciences, Nord University, Levanger, Norway. [19]Department of Biostatistics and Center for Statistical Genetics, University of Michigan, Ann Arbor, MI, USA. [20]Department of Internal Medicine, University of Michigan, Ann Arbor, MI, USA. [21]Department of Human Genetics, University of Michigan, Ann Arbor, MI, USA. [22]University of Queensland Diamantina Institute, University of Queensland,

Brisbane, QLD, Australia. [23]Department of Public Health, University of Helsinki, Helsinki, Finland. [24]Institute for Molecular Medicine Finland (FIMM), University of Helsinki, Helsinki, Finland. [25]Department of Endocrinology, St Olavs Hospital, Trondheim University Hospital, Trondheim, Norway. [26]These authors contributed equally: George Davey Smith, Bjørn Olav Åsvold, Gibran Hemani, Neil M. Davies. *Lists of authors and their affiliations appear at the end of the paper. ✉email: ben.brumpton@ntnu.no; neil.davies@bristol.ac.uk

## The Within-family Consortium

Ben Brumpton[1,2,3], Eleanor Sanderson[2,4], Karl Heilbron[5], Fernando Pires Hartwig[2,6], Sean Harrison[2,4], Gunnhild Åberge Vie[1], Yoonsu Cho[2,4], Laura D. Howe[2,4], Amanda Hughes[2,4], Dorret I. Boomsma[7], Alexandra Havdahl[2,8,9], John Hopper[10], Michael Neale[11], Michel G. Nivard[7], Nancy L. Pedersen[12], Chandra A. Reynolds[13], Elliot M. Tucker-Drob[14], Andrew Grotzinger[14], Laurence Howe[2,4], Tim Morris[2,4], Shuai Li[10,15], Adam Auton[5], Frank Windmeijer[2,16], Wei-Min Chen[17], Johan Håkon Bjørngaard[1,18], Kristian Hveem[1], Cristen Willer[19,20,21], David M. Evans[2,22], Jaakko Kaprio[23,24], George Davey Smith[2,4], Bjørn Olav Åsvold[1,25], Gibran Hemani[2,4] & Neil M. Davies[1,2,4]

## The 23andMe Research Team

Karl Heilbron[5] & Adam Auton[5]

A full list of consortium members is included in the Supplementary materials.

