## [Peer Review File · Nature Communications]

Review of “**Within-family studies for Mendelian randomization: avoiding dynastic, assortative mating, and population stratification biases**”

The authors examine the consequences of family effects in biasing the results from a Mendelian randomization (MR) analysis. The authors approach their analysis in two ways, first via a simulation study where they study the bias and power of standard MR analysis when the magnitude of the family effects is varied and second via an empirical study from the UK Biobank and Nord-Trøndelag Health Study that contains sibling information. Overall, they find that ignoring familial effects can pose a threat to standard MR analysis and whenever the data allows it, advocate using a within-family design to avoid biases arising from them.

To me, the authors’ work addresses an important question in MR analysis, which is the effect of family structure in estimating the desired exposure effect, and I agree with the authors’ general conclusions that family effects, if ignored, can bias MR analysis. The authors’ contribution to this not only alerting the MR community about these bias, but also showing how bad these biases can be in simulated and empirical studies.

Unfortunately, I wished that the authors provided some easy-to-use remedy to this important problem, or even a visual diagnostic to assess this problem. Beyond the general warning about these biases in MR which the authors quantitatively showed in their simulations, the only real cure to this from the authors’ empirical analysis is actually collecting sibling data. Since many MR analysis do not contain such information (or the analysis that do contain it have small sample sizes), the authors’ remedy of using within-family effects or differencing methods is not broadly applicable.

To this end, a room for improvement is to either (a) provide a sensitivity analysis that can be done with standard MR analysis to assess this bias, (b) propose a diagnostic plot/test, however imperfect, to detect these biases, or (c) suggest robust estimators if sibling information is not available. I think (c) is difficult without making strong assumptions about heritability and assortative mating behaviour. But, I think (a) and (b) are possible, especially (a) since family effects are essentially violations of the independence assumption and there are plethora of sensitivity analyses to deal with such violations. For example, for (a), I would either derive the bias of using the IVW estimator naively under the data generating model in (i) and (ii) and use the bias term as the range of the sensitivity parameter OR

1. Assume f_k exists, is known, and of a certain magnitude
2. Subtract it away from equations (i) and (ii), ending up with transformed outcomes/exposures,
3. Run the same analysis with the transformed outcomes/exposures
4. Repeat (1)-(3) with different structures on f_k , report the histogram of estimates from (3), and compare it to the estimate obtained by assuming no family effects.

The former proposal is more useful if only summary data is available and the latter proposal is useful if you have individual-level data.

For (b), I think you can use the over-identification test in econometrics since the instrument $g_{k,i}$ in equation (i) is acting like an endogenous variable when family effects not are considered.

Other comments are enumerated below.

1. Have the authors checked the magnitude and significance of the family effects in their data? I'm curious whether these were very large (or very small), especially in contrast to the actual exposure effect.

Also, if the family effect is significant in one regression (i.e. exposure-SNP), but not in the other regression (i.e. outcome-SNP), how should one interpret that, especially in the context of the DAGs A, B, and C in Figure 1?

2. Can the family effect dummy variable be a collider instead of a confounder? Currently, the DAGs in Figure 1, especially the direction of the arrows from the SNP to the outcome in the current sample, is treating the family effect as an unmeasured confounder. But, it is possible that the family effect dummy variable contains other information about family structure that comes (temporally) after the SNP variable in the current sample's DAG, and this may ultimately influence the outcome? In short, would the family effect dummy variable also encode post-treatment conditions that may bias the exposure effect?
3. Are the biases from family effects amplified in the presence of weak instruments? One would imagine that if the $g_{k,i}$ in equation 1 is weakly correlated with the exposure and if $g_{k,i}$ is strongly correlated with $f_{k,i}$, the bias would be amplified because the instrument's endogeneity to the error term is larger than its effects on the exposure.
4. I think there is a typo in the error terms for equation (iv) and (v). The $u_{k,j}$ is not the same as the $u_{k,i}$ in equation (ii). Additionally, the errors in equation (iv) and (v) cannot have mean zero because of the squaring. The Appendix actually uses more precise and correct notation for this.

Reviewer #2 (Remarks to the Author):

In this manuscript, the authors pointed out that performing mendelian randomization using 'unrelated' samples is subject to various forms of bias when assumptions are violated, and much of this bias could be reduced/eliminated by using within-family analyses, e.g. parents-offspring trios or siblings. In addition to some simulation results, empirical analyses/examples highlighting the difference between analyses performed using 'unrelated' probands and using siblings are presented.

I agree with the general premise of the paper, and find the empirical examples interesting. However, I believe the presentation of the material, particular the technical aspects, could be substantially improved.

Specific Comments

1. Power investigation, Fig. 2. Power is an extremely non-linear function of effect magnitude and sample size. If at all possible, results should also be presented based on standard errors, exact or relative, e.g. how many sibpairs is needed to have (approximately) the same standard errors as n unrelated samples. Power can be determined by effect, bias and magnitude of the standard errors. I do not object to displaying some power curves directly, but some theoretical calculations would be useful. In panel A of Fig 2, the x-axis range from 0.00 to 0.05, but simulations are only performed for a few x-values, and points joined by straight lines. So they are not really 'curves'. Furthermore, the points often do not look like they fall on a smooth curve. For example, with Dynastic = 0.2, the results for Sibling look particularly odd, the slope decreases, then increases, and decreases again. Can that be real? In panel B. Even the Sibling and Trio MR have substantial biases when sample size is modest and decreases when sample size increases. The question is whether the estimate is supposed to asymptotically unbiased (under the simulation model)? If not, why? If the estimate is asymptotically unbiased, is the finite/small sample bias a result of the estimate being a ratio-estimate? If so, would it be helpful to evaluate the estimate in the log scale?
2. Simulations and Fig 2 continued. For panel A, sample size is 10000, does that mean 10000 trios and 5000 sibpairs? Lines 163-164, C_X, C_Y and beta_ux, are not defined. In general, the authors should make sure that things are defined when they first appear in the text.
3. Empirical study. The empirical study is the most important part of this manuscript, but it only takes up 68 lines in the main text, while other parts of the paper, e.g. Discussion, are much longer. Details on exactly how the analyses are performed, which are important, are mostly missing. I appreciate that much of the details are in Methods, but even there, they are not organized in a way that makes it easy for the reader. For example, the section Empirical analyses (line 646 to line 669) is rather brief and without pointing to other parts of Methods. Covariates and standard errors (lines 635-640), which appear earlier, and with Ethics in between, describe PCs adjustment etc. are an important part of the actual analyses. Selections of genotypes for instruments (lines 550 to 556) are also specific to the analyses performed. The statistical models (lines 403 to 477) are also key to the analyses I presume. By putting all of this together, I can guess how the empirical results are produced. But it should not take so much effort on the part of reader, and it should not require guessing.
4. Dynastic effects and sib effects. In Box 1 of Supplementary material (page 44), the description of Dynastic effects highlight the effects of parents, but I presume that it can also include effects from other blood relatives such as grandparents, and maybe even uncle/aunts and cousins? The authors should clarify that. One effect is particularly relevant technically. If genotypes of a sibling, through his/her phenotypes, can impact the outcomes of the proband, what effect would it have on the estimates coming from sibling MR? That has to be addressed. The paragraph from line 309 to 319 might be related to this issue. But something more quantitatively explicit would help.
5. Lines 234 to 235. The authors suggest that the effects of height on education could be due to dynastic effects or assortative mating. Dynastic effect is a true genetic effect even though it manifests indirectly through phenotypes of blood relatives. While distinct, it is not less important

than direct genetic effect. The question is how could the height of parents have a (indirect) causal effect on the education of the child. Line 275 to line 278 might be related to that. If so, should be more explicit. With assortative mating, the effect I am most aware of is a confounding effect that magnifies the observed effect of a causal variant because of correlations with other causal variants. But this would not generate a positive effect from zero.

6. PC adjustment. Why is principal component adjustment needed for within family analyses? Shouldn't the latter by itself take care of the confounding that PC adjustment is meant to do?

Reviewer 1

Review of “Within-family studies for Mendelian randomization: avoiding dynastic, assortative mating, and population stratification biases”

The authors examine the consequences of family effects in biasing the results from a Mendelian randomization (MR) analysis. The authors approach their analysis in two ways, first via a simulation study where they study the bias and power of standard MR analysis when the magnitude of the family effects is varied and second via an empirical study from the UK Biobank and Nord-Trøndelag Health Study that contains sibling information. Overall, they find that ignoring familial effects can pose a threat to standard MR analysis and whenever the data allows it, advocate using a within-family design to avoid biases arising from them.

To me, the authors’ work addresses an important question in MR analysis, which is the effect of family structure in estimating the desired exposure effect, and I agree with the authors’ general conclusions that family effects, if ignored, can bias MR analysis. The authors’ contribution to this is not only alerting the MR community about these biases, but also showing how bad these biases can be in simulated and empirical studies.

Unfortunately, I wished that the authors provided some easy-to-use remedy to this important problem, or even a visual diagnostic to assess this problem. Beyond the general warning about these biases in MR which the authors quantitatively showed in their simulations, the only real cure to this from the authors’ empirical analysis is actually collecting sibling data. Since many MR analysis do not contain such information (or the analysis that do contain it have small sample sizes), the authors’ remedy of using within-family effects or differencing methods is not broadly applicable.

To this end, a room for improvement is to either (a) provide a sensitivity analysis that can be done with standard MR analysis to assess this bias, (b) propose a diagnostic plot/test, however imperfect, to detect these biases, or (c) suggest robust estimators if sibling information is not available. I think (c) is difficult without making strong assumptions about heritability and assortative mating behaviour. But, I think (a) and (b) are possible, especially (a) since family effects are essentially violations of the independence assumption and there are plethora of sensitivity analyses to deal with such violations. For example, for (a), I would either derive the bias of using the IVW estimator naively under the data generating model in (i) and (ii) and use the bias term as the range of the sensitivity parameter OR

1. Assume f_k exists, is known, and of a certain magnitude
2. Subtract it away from equations (i) and (ii), ending up with transformed outcomes/exposures,
3. Run the same analysis with the transformed outcomes/exposures
4. Repeat (1)-(3) with different structures on f_k , report the histogram of estimates from (3), and compare it to the estimate obtained by assuming no family effects.

The former proposal is more useful if only summary data is available and the latter proposal is useful if you have individual-level data.

For (b), I think you can use the over-identification test in econometrics since the instrument $g_{\{k,i\}}$ in equation (i) is acting like an endogenous variable when family effects not are considered.

We describe how samples of related individuals can be used to detect familial biases in MR. If there are differences between estimates from unrelated individuals and related individuals allowing for a familial effect, then this suggests that the former estimates are

biased. We agree that methods or sensitivity analyses to assess these biases using estimates from samples of unrelated individuals would be useful. However, to our knowledge there are no currently available methods for assessing the extent of these biases without data from related individuals. Most existing methods have focused on overcoming bias due to violations of the exclusion restriction assumption (i.e. horizontal pleiotropy) – e.g. MR-Egger, weighted median and weighted mode.(1–3) However, familial effects will induce bias that is proportional to the effect of the SNP on the exposure of interest, this will violate the InSIDE assumption. Similarly, this means that the median or modal estimate will also be biased. The Genetic Instrumental Variables (GIV) approach can be robust to these biases, but also requires samples of related individuals.(4)

The reviewer suggests running sensitivity analyses that assume a given strength of familial confounding to see if it can induce the magnitude of results found. The analytical bias terms for the dynastic effects are a form of omitted variable bias and are derived below. We have added this to the supplementary materials.

“Analytic derivation of familial sources of bias as omitted variable bias

Dynastic effects (genetic nurture)

Consider two parents, a father and mother, f and m who at a specific bivariate locus have one of two alleles of a given frequency:

$$g_{1f} \sim \text{Bernonlli}(p) \quad (\text{i})$$

$$g_{2f} \sim \text{Bernonlli}(p)$$

$$g_{1m} \sim \text{Bernonlli}(p)$$

$$g_{2m} \sim \text{Bernonlli}(p)$$

Thus, at each locus, the parents can be one of three genotypes, homozygous 00, homozygous 11, or heterozygous 10. Each pair of parents have one offspring, which inherits one of each of the parent’s two alleles. The offspring inherit their alleles at random from their parents as indicated by the variable $t_m, t_f \sim \text{Bernonlli}(0.5)$:

$$g_{1i} = t g_{1f} + (1 - t) g_{1m} \quad (\text{ii})$$

$$g_{2i} = (1 - t) g_{2f} + t g_{2m}$$

The parents’ phenotypes are a function of their genotypes and a random environmental error term:

$$p_f = \beta(g_{1f} + g_{2f}) + e_f \quad (\text{iii})$$

$$p_m = \beta(g_{1m} + g_{2m}) + e_m$$

Where β is a constant, the paternal and maternal independent error terms are indicated by e_f and $e_m \sim N(\mu, \sigma^2)$, and constants are not shown. Similarly, the offspring phenotype function of the offspring genotype and an independent environmental error term. In addition, the offspring phenotype can also be affected by a dynastic, or genetic nurturing effect of the parental phenotype on the outcome indicated by γ :

$$p_i = \beta(g_{1i} + g_{2i}) + \gamma(p_f + p_m) + e_i \quad (\text{iv})$$

The maternal, paternal and offspring genotypes are $g_m = g_{1m} + g_{2m}$, $g_f = g_{1f} + g_{2f}$ and $g_i = g_{1i} + g_{2i}$ respectively, and $e_i \sim N(\mu, \sigma^2)$.

The OLS estimator for β from regressing the offspring phenotype on the offspring genotype is the equation for omitted variable bias(5):

$$\hat{\beta}_{ols} = \beta + \frac{\sum g_i (\gamma(p_f + p_m) + e_i)}{\sum (g_i)^2} \quad (\text{v})$$

The error term e_i is independent of g_i so this can be simplified to:

$$\hat{\beta}_{ols} = \beta + \frac{\gamma \sum g_i (p_f + p_m)}{\sum (g_i)^2} \quad (\text{vi})$$

Substituting in for parental phenotypes:

$$\hat{\beta}_{ols} = \beta + \frac{\gamma \sum g_i (\beta g_f + e_f + \beta g_m + e_m)}{\sum (g_i)^2} \quad (\text{vii})$$

The error terms e_f and e_m are independent of g_i so this can be simplified to:

$$\hat{\beta}_{ols} = \beta + \frac{\gamma \beta \sum g_i (g_f + g_m)}{\sum (g_i)^2} \quad (\text{viii})$$

The covariance between parental and offspring genotypes is equal to half the variance of the genotype, therefore:

$$\hat{\beta}_{ols} = \beta(1 + \gamma) \quad (\text{ix})$$

The bias in the OLS estimate will be:

$$\text{bias}(\hat{\beta}_{ols}) = \beta\gamma \quad (x)$$

This implies that the bias in the OLS estimate will be the product of the dynastic effect and the true causal effect of the genotype on the phenotype in the outcome.”

The bias due to assortative mating can also be derived as follows – we have added this to the results section of the paper:

“Less intuitive is that some patterns of assortative mating can induce SNP-outcome relationships that are due to bias, in that they do not arise due to a counterfactual allelic substitution occurring at the individual level. This is problematic for GWAS in general, but MR specifically if the SNP in question is being used to instrument the exposure under analysis. This has been examined through simulation in detail before, and here we examine how the bias arises from a theoretical perspective.(6) Let ρ be the correlation between the male exposure phenotype x_m and the female outcome phenotype y_f that arises due to assortment. If x causes y , then any genetic influence on x is also an influence on y , but for bias to arise we are interested in the possibility of there being an association between y and the instruments for x that is not through a biological causal relationship. To this end, suppose x and y are each heritable, and we have an instrument g_x for x , let us define a genetic score for y , s_y , that is biologically independent of g_x . The system being analysed is:

$$\begin{aligned} x_i &= \gamma_i g_{x,i} + C_i + u_i \\ y_i &= \beta_i x_i + s_y + C_i + v_i \end{aligned} \quad (i)$$

where C is a confounder, u and v are error terms, and $i \in \{m, f, o\}$ is used to denote the group of individuals in which the parameters are being estimated – males, females, offspring. The causal effect estimate is obtained as

$$\hat{\beta}_i = \frac{\text{cov}(g_{x,i}, y_i)}{\text{cov}(g_{x,i}, x_i)} \quad \text{(ii)}$$

We know that due to Mendelian inheritance $\text{cor}(g_m, g_o) = \text{cor}(g_f, g_o) = 0.5$. We can also infer that following assortment of male x and female y phenotypes, the expected covariance between their respective genetic factors will be

$$\text{cov}(g_{x,m}, s_{y,f}) = \rho \cdot \text{cov}(g_{x,m}, x_m) \cdot \text{cov}(s_{y,f}, y_f) \quad \text{(iii)}$$

The assortative mating induces a covariance between the genetic instrument for x and the genetic score for y that is biologically independent of x:

$$\begin{aligned} \text{cov}(g_{x,o}, s_{y,o}) & \quad \text{(iv)} \\ &= \text{cov}(g_{x,m}, s_{y,f}) \cdot \text{cor}(s_{y,f}, s_{y,o}) \\ & \quad \cdot \text{cor}(g_{x,m}, g_{x,o}) \\ &= \frac{1}{4} \cdot \rho \cdot \text{cov}(g_x, x) \cdot \text{cov}(s_y, y) \end{aligned}$$

Given that there is a direct biological influence of $s_{y,o}$ on y_o , substituting back into the causal effect estimate we find that if $\rho \neq 0$ then the Mendelian randomization estimate will be biased:

$$\begin{aligned} \hat{\beta}_o &= \frac{\text{cov}(g_{x,o}, y_o) + \frac{\text{cov}(g_{x,o}, s_{y,o})}{\text{cor}(y_o, s_{y,o})^2}}{\text{cov}(g_{x,o}, x_o)} \quad \text{(v)} \\ &= \frac{\text{cov}(g_{x,o}, y_o) + \frac{1}{4} \cdot \rho \cdot \text{cov}(g_x, x) \cdot \text{cov}(s_y, y)}{\text{cov}(g_{x,o}, x_o)} \end{aligned}$$

As this has been shown in simulation before,⁽⁶⁾ we now continue by demonstrating the utility of within-family designs for protecting MR estimates from bias due to family structures, and then illustrate their importance using empirical examples.”

Other comments are enumerated below.

1. Have the authors checked the magnitude and significance of the family effects in their data? I’m curious whether these were very large (or very small), especially in contrast to the actual exposure effect.

The analysis did not include a within-family effect (i.e. include the familial level mean as a covariate – method), as we used family fixed effects estimators.

In addition, we estimated the shrinkage of the total to within-family estimates of the SNP-phenotype associations using Seemly Unrelated Regression (SUR). SUR allows us to estimate the ratio of the within-family to total associations using the same sample and account for the covariance between the estimates. We estimated the ratios of the within-family to total SNP-phenotype association for the 455 SNPs included in the analysis and the five phenotypes (education, BMI, height, diabetes and high blood pressure). We then meta-analysed estimates for each SNP and phenotype across the two studies. Finally, we meta-analysed across all 455 SNPs used in the analysis to give an average shrinkage for all SNPs. The estimated ratios are given below in Table 1. This shrinkage of the estimates suggests that allowing for a family fixed effects affects the estimated SNP-phenotype associations for all phenotypes.

We have clarified the estimators we use in the paper in the methods and have included the following table which reports the shrinkage in UK Biobank and HUNT. We have added the following text to the methods:

“An analytically convenient method to use for this estimator is the within transformation. The within transformation either demeans the variables for or adjusts for the family level means. Demeaned using the within transformation is computationally efficient, particularly for large sample sizes – and is the analytic method used by many statistical packages for fixed effects estimators. An advantage of adjusting for the within-family mean, rather than demeaning is that it provides an estimate of the between family effect.”

We added a paragraph in the results describing the shrinkage analysis:

“We investigated the difference (shrinkage) of the total to within-family SNP-phenotype associations in HUNT and UK Biobank using seemly unrelated regression (SUR). The estimated shrinkage is given in Supplementary Table 1. The shrinkage of the estimates suggests that allowing for a family fixed effects affects the estimated SNP-phenotype associations for all phenotypes. Educational attainment was the most strongly affected, falling by 56.8% (95%CI: 49.2% to 64.4%). Diabetes was the least affected falling by 11.2% (95%CI: 1.3% to 21.1%).”

We added the following text to the methods to describe the shrinkage analysis:

“Shrinkage

We investigated shrinkage of the total to within-family SNP-phenotype associations using seemingly unrelated regression. We estimated the shrinkage for each of the 455 SNPs included in the analysis and the five phenotypes (education, BMI, height, diabetes and high blood pressure). We then meta-analysed estimates for each SNP and phenotype across the two studies. Finally, independently for each phenotype we meta-analysed across all 455 SNPs used in the analysis to give an average shrinkage for all SNPs.”

Supplementary Table 1: The shrinkage of the within-family SNP-phenotype associations. Estimated in UK Biobank and HUNT using seemingly unrelated regression.

Phenotype	Ratio	95% Confidence interval
Education	0.432	0.356 to 0.508
BMI	0.821	0.760 to 0.881
Height	0.749	0.721 to 0.777
High blood pressure	0.721	0.627 to 0.814
Diabetes	0.888	0.789 to 0.987

Also, if the family effect is significant in one regression (i.e. exposure-SNP), but not in the other regression (i.e. outcome-SNP), how should one interpret that, especially in the context of the DAGs A, B, and C in Figure 1?

We will split our response into theoretical and empirical responses to this question.

Theoretical

If there are familial effects on the exposure, but there are no familial effects on the outcome, then the Mendelian randomization estimate will be unbiased under the null. This means it will be a valid test of the null hypothesis that the exposure causes the outcome. To see why consider the following DAGs, which have been modified from **Figure 1b** to remove the familial effects on the outcome (Figure 1):

Figure 1. A directed acyclic graph (DAG) illustrating the relationships and potential confounding mechanisms if the familial effects do not affect the outcome. In each case, there are no open paths from the SNP to the outcome, therefore familial effects that are mediated solely through the exposure are unlikely to cause bias.

Population stratification which confounds only the SNP-exposure association will accordingly bias the SNP-exposure association but not bias in the SNP-outcome association. Absent other violations of the MR assumptions, the overall MR estimate, which is a ratio of SNP-outcome and SNP-exposure associations will be an unbiased estimate of the causal effect. This is because the SNP-exposure association is confounded by population structure, but this confounding only affects the level of the exposure, which then has a downstream effect (or not) on the outcome. Similarly, if there are dynastic effects from parent's exposure genotype to offspring exposure phenotype, then the MR estimate will again be unbiased under the null of no effect of the exposure on the outcome and will provide unbiased estimates of the effect of the exposure on the outcome. This is because the dynastic effects affect the level of

the exposure. These effects mean the SNP-exposure association is no longer an unbiased estimate of manipulating the SNP in the offspring – it also reflects the effect of the SNP via the parent. However, if the dynastic effects only influence the exposure, and have no other effects on the outcome, then the instrumental variable assumptions are not violated. In the Figure below, this is indicated by the absence of a confounding path between the SNP and the outcome. Finally, single trait assortative mating on the exposure will induce correlations between phenotypic exposure and SNPs for the exposure. As a result, the SNP-exposure association will again be a biased estimate of the causal effect of changing that SNP in the offspring. However, there are no paths between the SNPs for the exposure and the outcome, except via the exposure, therefore the MR estimate will be unbiased under the null of no effect and provide unbiased estimates of the effect of the exposure on the outcome.

Empirical

Therefore, if there is evidence of a familial effect in the SNP-exposure, but not in the SNP-outcome association and these estimates are well powered, then the MR estimates using unrelated individuals are also likely to be unbiased. The estimates from unrelated individuals should be preferred because they are likely to be more precise. The within-family MR estimates are less likely to be biased by familial effects but more imprecise than the MR estimates from unrelated individuals. As a result, the within-family estimates could be considered a form of sensitivity analysis, which is more accurate, but less precise.

We have added Figure 1 (as Supplementary Figure 3 in the manuscript) and the following text in the results section to cover this point.

“If the familial effect affects the exposure, but does not affect the outcome, then we would not expect bias in the Mendelian randomization analysis. This is because there would be no open path between the SNP and the outcome. This is illustrated in the directed acyclic graph illustrated in **Supplementary Figure 3.**”

2. Can the family effect dummy variable be a collider instead of a confounder? Currently, the DAGs in Figure 1, especially the direction of the arrows from the SNP to the outcome in the current sample, is treating the family effect as an unmeasured confounder. But, it is possible that the family effect dummy variable contains other information about family structure that comes (temporally) after the SNP variable in the current sample’s DAG, and this may ultimately influence the outcome? In short, would the family effect dummy variable also encode post-treatment conditions that may bias the exposure effect?

In phenotypic family-based designs it is possible for the family environment to be a collider of the association of an exposure and an outcome after conditioning on a familial effect.(7) This can occur if both the exposure and outcome affect the family environment. Sjölander and Zetterqvist (2017) discuss the example of maternal age at birth and offspring ADHD risk. They argue that maternal age at birth and having a sibling with ADHD are likely to affect the shared familial environment, and thus shared familial environment is a collider of maternal age and offspring ADHD. However, our analyses used genotype to estimate the causal effect of an exposure on outcome (Figure 2). The within-family MR estimates use the SNP-exposure and SNP-outcome associations. In contrast to phenotypic within-family estimates discussed by Sjölander and Zetterqvist (2017), in general, genotype is invariant after conception. Furthermore, each offspring conception is a random draw from their parents’ genotypes,

and thus which genotype each sibling inherits will not affect the genotype received by their siblings. These facts mean that it is impossible for the family environment to be a collider of the SNP-phenotype association.

Figure 2. DAGs illustrating the expected bias if the familial effects are colliders of the SNP-phenotype association. Theoretically, if both the SNP and the outcome cause each of the forms of familial effect, then there would be an open path from the SNP to the outcome, and the MR estimate would be biased. However, in example A, it is impossible for the offspring's SNP to affect the population demography and structure in the population. Similarly, it is impossible for the offspring SNP to affect their parents' SNPs (i.e. reverse dynastic effects). In example C, parental cross trait assortment could be illustrated as a collider, as selection on the parental exposure and outcome phenotypes. This would represent an open path from offspring exposure SNP to the outcome and would cause bias. We have updated the DAGs to illustrate this more accurately as a collider in the manuscript. However, conditional on parental (or sibling) genotype, this path would be closed. Thus, each case is either unlikely, impossible or likely to be controlled in a within-family analysis.

3. Are the biases from family effects amplified in the presence of weak instruments? One would imagine that if the $g_{\{k\}}$ in equation 1 is weakly correlated with the exposure and if $g_{\{k\}}$ is strongly

correlated with $f_{\{k\}}$, the bias would be amplified because the instrument's endogeneity to the error term is larger than its effects on the exposure.

In short yes, in samples of unrelated individuals, biases from familial effects will be amplified if the genetic variants only have a small causal effect on the exposure. A well-known feature of Wald type instrumental variable estimators (i.e. estimators that comprise a ratio of the instrument-outcome and instrument-exposure associations), is that any bias in the numerator (the instrument-outcome association) will be magnified if the denominator (the instrument-exposure association) is small. However, the consequences of familial biases for MR are not just a function of the strength of the instruments and depend on the specifics of an analysis. In samples of unrelated individuals which cannot account for these biases, using weaker instruments is likely to lead to a larger bias. I.e. holding the size of the familial effects constant, if the causal effect of the SNP on the exposure falls, the bias will increase. However, in samples of related individuals and using models which allow for familial effects, the SNP-phenotype associations are less likely to be biased by population structure, dynastic effects or assortative mating. We have edited the first paragraph of the discussion to make this point:

"We have presented within-family methods for Mendelian randomization and demonstrated how confounding due to family structure can bias Mendelian randomization studies using unrelated individuals. As with most instrumental variable estimators, *ceteris paribus* the size of the bias induced by familial effects will be larger the smaller the causal effect of the genetic variant on the exposure."

4. I think there is a typo in the error terms for equation (iv) and (v). The $u_{\{k,j\}}$ is not the same as the $u_{\{k,j\}}$ in equation (ii). Additionally, the errors in equation (iv) and (v) cannot have mean zero because of the squaring. The Appendix actually uses more precise and correct notation for this.

We have corrected this notation in the revised manuscript.

Reviewer 2

In this manuscript, the authors pointed out that performing mendelian randomization using 'unrelated' samples is subject to various forms of bias when assumptions are violated, and much of this bias could be reduced/eliminated by using within-family analyses, e.g. parents-offspring trios or siblings. In addition to some simulation results, empirical analyses/examples highlighting the difference between analyses performed using 'unrelated' probands and using siblings are presented.

I agree with the general premise of the paper and find the empirical examples interesting. However, I believe the presentation of the material, particular the technical aspects, could be substantially improved.

Specific Comments

1. Power investigation, Fig. 2. Power is an extremely non-linear function of effect magnitude and sample size. If at all possible, results should also be presented based on standard errors, exact or

relative, e.g. how many sibpairs is needed to have (approximately) the same standard errors as n unrelated samples.

Power can be determined by effect, bias and magnitude of the standard errors. I do not object to displaying some power curves directly, but some theoretical calculations would be useful.

Our empirical example also provides some indication of power. Directly comparing the standard errors of the MR estimates with and without including family fixed effects suggests that we would need 2.1 and 2.94 times as many siblings (corresponding to approximately the same number of sibling pairs as unrelated individuals) to achieve a similar level of precision (Table 2). The within-family two sample MR estimates were slightly more precise and suggested that the we would need 1.50 to 2.94 as many siblings as singletons.

Table 2. Ratio of the standard MR and within-family MR standard errors for each of the empirical examples presented in Figure 3.

Phenotype	Ratio of SEs of MR-PRS family FE/MR-PRS siblings (increase in sample size required)	Ratio of SEs of 2SMR IVW siblings – split sample/MR-PRS siblings (increase in sample size required)
BMI on diabetes	1.71 (2.94)	1.50 (2.24)
BMI on high blood pressure	1.71 (2.93)	1.49 (2.22)
Height on education	1.57 (2.48)	1.39 (1.94)
BMI on education	1.45 (2.10)	1.23 (1.50)

We have added this table to the supplement (as Supplementary Table 2 in the manuscript) and a description of this analysis to the discussion as follows:

“Supplementary Table 2 contains the ratio of standard errors for the MR analyses using unrelated individuals and allowing for a familial effect for the empirical results for the MR-PRS and 2SMR IVW results. These estimates use identical samples, but the within-family estimates had standard errors that were between 23% and 71% larger. This implies that the within-family analyses would require total samples sizes between 150% and 294% larger.”

In panel A of Fig 2, the x-axis ranges from 0.00 to 0.05, but simulations are only performed for a few x-values, and points joined by straight lines. So, they are not really ‘curves’. Furthermore, the points often do not look like they fall on a smooth curve. For example, with Dynastic = 0.2, the results for Sibling look particularly odd, the slope decreases, then increases, and decreases again. Can that be real?

We have increased the number of scenarios in the range between 0 and 0.05 % variance explained by the instruments and increased the number of simulations per scenario from 100 to 1000 to get less noisy estimates, which is what was explaining the fluctuations.

Figure 2: Updated figures illustrating bias and power for within-family methods using siblings and trios.

In panel B. Even the Sibling and Trio MR have substantial biases when sample size is modest and decreases when sample size increases. The question is whether the estimate is supposed to asymptotically unbiased (under the simulation model)? If not, why? If the estimate is asymptotically unbiased, is the finite/small sample bias a result of the estimate being a ratio-estimate? If so, would it be helpful to evaluate the estimate in the log scale?

The simulations indicate weak instrument bias towards the observational estimate due to sample overlap between the exposure and the outcome in the simulations. The bias decreases as the variance explained by the instruments increases. There are several processes that can simultaneously introduce bias to MR, and while the focus of this study is on population dynamics, care should be taken in empirical studies to avoid other forms of bias such as weak instrument or sample overlap bias.

2. Simulations and Fig 2 continued. For panel A, sample size is 10000, does that mean 10000 trios and 5000 sibpairs? Lines 163-164, C_X, C_Y and beta_ux, are not defined. In general, the authors should make sure that things are defined when they first appear in the text.

We have clarified this point. The sample sizes represent the number of points of information, so for the standard MR it is the number of samples, for the sibling analysis it is the number of siblings, and the trios is the number of trios. We have edited the legend of this figure to read:

“Figure 2: Estimated false discovery rate by power of the studies using different Mendelian randomization designs. A: SNP-exposure $r^2 = 0.05$; sample size = 10000 singletons, siblings, or trios; simulation involves an influence of parental exposure influencing child’s confounder, which explains 10% of variance in child exposures and outcomes. For a simulated causal effect = 0, we expect the false discovery rate to be 0.05. B: Estimated bias by sample size using different Mendelian Randomization designs. The simulations are similar to panel (A) but allow sample size to vary and fixing the causal effect of an exposure x on an outcome y to 1% of variance explained. The bias in within-family Mendelian randomization estimates is small unless the dynastic effects are very small, or the number of observations modest.”

Empirical study. The empirical study is the most important part of this manuscript, but it only takes up 68 lines in the main text, while other parts of the paper, e.g. Discussion, are much longer. Details on exactly how the analyses are performed, which are important, are mostly missing. I appreciate that much of the details are in Methods, but even there, they are not organized in a way that makes it easy for the reader. For example, the section Empirical analyses (line 646 to line 669) is rather brief and without pointing to other parts of Methods. Covariates and standard errors (lines 635-640), which appear earlier, and with Ethics in between, describe PCs adjustment etc. are an important part of the actual analyses. Selections of genotypes for instruments (lines 550 to 556) are also specific to the analyses performed. The statistical models (lines 403 to 477) are also key to the analyses I presume. By putting all of this together, I can guess how the empirical results are produced. But it should not take so much effort on the part of reader, and it should not require guessing.

We have revised the methods section as suggested, including reordering the material and adding additional detail into the empirical analyses section (pages 23 to 31).

3. Dynastic effects and sib effects. In Box 1 of Supplementary material (page 44), the description of Dynastic effects highlights the effects of parents, but I presume that it can also include effects from other blood relatives such as grandparents, and maybe even uncle/aunts and cousins? The authors should clarify that. One effect is particularly relevant technically. If genotypes of a sibling, through his/her phenotypes, can impact the outcomes of the proband, what effect would it have on the estimates coming from sibling MR? That has to be addressed. The paragraph from line 309 to 319 might be related to this issue. But something more quantitatively explicit would help.

We have updated Box 1 to include effects of more distance relatives within the definition of dynastic effects:

“**Dynastic effects (genetic nurture)**: when parental genotype affects offspring outcomes through pathways other than via offspring genotype. For example, if more educated parents support their offspring’s education, or if parents smoking positively or negatively affected the likelihood of their offspring smoking. An example of dynastic effects are passive gene-environmental correlations.^(8–10) Other relationships, such as grandparents, uncles/aunts and cousins may affect the offspring’s phenotype – these can also be thought of as a form of dynastic effect.”

Sibling effects could bias MR estimates of the effects of an exposure on an outcome using data from unrelated individuals (Figure 3). For example, suppose that taller siblings were more likely to encourage their siblings to engage in sports rather than academic activities. This would mean that people with siblings with more height associated variants, were more likely to be encouraged by their siblings to play sport, which may reduce their academic attainment. In which case, individuals with more height variants, who are more likely to have taller siblings, would have a slightly lower academic attainment. However, within a family it is not possible for siblings to affect their brother or sister's genotype. Genotype is determined at conception by parental genotype and chance. Therefore, sibling effects are unlikely to confound within-family MR estimates, however family methods should be extended to quantify this effect. We have now included a statement in the introduction (page 1 paragraph two) specifically referring to sibling effects and other relationships that might induce bias in the MR of unrelated individuals.

Figure 3: Figure illustrating sibling effects and how this pathway is blocked by conditioning on parental genotype.

4. Lines 234 to 235. The authors suggest that the effects of height on education could be due to dynastic effects or assortative mating. Dynastic effect is a true genetic effect even though it manifests indirectly through phenotypes of blood relatives. While distinct, it is not less important than direct genetic effect. The question is how the height of parents could have a (indirect) causal effect on the education of the child. Line 275 to line 278 might be related to that. If so, should be more explicit. With assortative mating, the effect I am most aware of is a confounding effect that magnifies the observed effect of a causal variant because of correlations with other causal variants. But this would not generate a positive effect from zero.

Absolutely, dynastic effects are an extremely interesting form of genetic effect. A form of indirect genetic effect that could affect height are variants involved in metabolism or nutrition. Suppose there was a variant that affects the parent's food consumption (e.g. variants related to anorexia nervosa). This could affect both the parents' height, and their offspring's neurodevelopment and educational attainment. However, dynastic effects are unlikely to explain these associations. We have modified the statements in the results to refer to familial effects, and modified the section in the discussion as follows:

In the results:

“These results suggest that the estimates in unrelated individuals may be due to familial effects.”

In the discussion:

“Consistent with our results, previous studies using twin data have indicated that the relationship between height and educational attainment is likely to be due to familial effects. (11, 12) These findings raise questions about whether height and BMI have individual level causal effects on socioeconomic outcomes later in life.^{49–51} These results suggest that familial effects can have important phenotypic consequences on widely studied relationships such as between height and BMI and education.”

Single trait assortative mating can inflate a SNP-phenotype association, because a causal variant would correlate with other, genomically distal, variants for that phenotype. This association will provide a valid test of the null hypothesis that the variant has no effect on the phenotype. This is because if the variant does not affect the phenotype then it will not be affected by assortative mating on that phenotype. However, perhaps the most likely source of bias in a MR estimate of the effect of height on educational attainment is from cross-trait assortative mating. This can occur when taller people assort with more educated individuals. The MR estimate will be biased even under the null hypothesis that height does not affect educational attainment. This is because if individuals assort on height and education, then those that inherit more height variants and also more likely to inherit variants associated with educational attainment. As a result, genetic variants associated with height will also associate with educational attainment, via the effects of other SNPs. Thus, cross-trait assortative mating can induce bias in MR studies using unrelated individuals. However, within families genotype will be inherited independently. Thus, studies using related individuals, such as siblings or parent-offspring trios, will be robust to this source of bias. We have clarified this point in the discussion as follows:

We have clarified this point in the second paragraph of the discussion:

“Population structure and dynastic effects can cause bias under the null hypothesis of no effect – i.e. induce spurious false positive signals. Whereas assortative mating on a single phenotype will be unbiased under the null hypothesis that the SNP does not affect the phenotype. Single trait assortative mating can inflate SNP-phenotype associations. However, cross trait assortative mating can cause bias under the null.”

5. PC adjustment. Why is principal component adjustment needed for within family analyses? Shouldn't the latter by itself take care of the confounding that PC adjustment is meant to do?

As indicated by Figure 1b, after conditioning on genotype, either via parental or siblings, ancestry cannot confound the SNP-phenotype association. Therefore, ancestry cannot cause bias in a SNP-phenotype association within a family. There are two reasons to include the principal components. First, the principal components may absorb a fraction of the variation in the outcome and thus increase precision (e.g. if there are regional differences in rates of the phenotype). Second, if one is interested in the between family estimates, then it can be helpful to include the PCs. Adjusting for PCs means that the between family estimates can be interpreted as likely to be due to familial level factors rather than major differences in population structure. We have noted the first point in the methods section, the second point is perhaps less relevant for this paper as we do not focus on estimating the familial effects:

"Covariates and standard errors

HUNT and UK Biobank analyses included age, sex, the first 20 principal components of genetic variation. Cluster robust standard errors were used to allow for heteroskedasticity and allow for clustering and relatedness across siblings within families. Inclusion of the covariates age, sex, and principal components did not meaningfully affect the within-family estimates, as they are independent of genotype conditional on sibling genotype. However, including these covariates may absorb some of the variation in the outcome and increase the precision of our estimates."

REVIEWERS' COMMENTS:

Reviewer #1 (Remarks to the Author):

The authors have addressed my questions. The approach is important in settings where large family data is available and I applaud the authors for addressing the importance of controlling hidden biases by adjusting for confounders related to family structure.

I have a few minor comments about the analytic strategy:

1) Under the split-sample approach, there is an inherent assumption that the family structures would be similar across both samples. While this is also a concern for typical MR studies, in this context, the split-sample approach assumes that within-family effects (and biases) are assumed to be similar between two samples and I was curious what the authors' thoughts are on this issue.

2) The analytical models assume no confounding due to interaction between family structure and sibling-specific characteristics. For example, even after removing for family effects, residual confounding may still persist where this residual confounding is a product of family effects and individual-level characteristics. I think some light discussion on this matter could be useful.

3) Is there a possibility that horizontal pleiotropy may reduce the effects due to confounding by family structure in your analytical models? I realize this concern is mostly theoretical, but I was curious if the authors have put some thought into this possibility.

Reviewer #2 made no further comments to the authors.

Reviewer comments

The authors have addressed my questions. The approach is important in settings where large family data is available and I applaud the authors for addressing the importance of controlling hidden biases by adjusting for confounders related to family structure.

I have a few minor comments about the analytic strategy:

1) Under the split-sample approach, there is an inherent assumption that the family structures would be similar across both samples. While this is also a concern for typical MR studies, in this context, the split-sample approach assumes that within-family effects (and biases) are assumed to be similar between two samples and I was curious what the authors' thoughts are on this issue.

The split sample does not require the familial effects to be the same in the two samples, it only requires the individual level causal effects to be the same. Suppose there was strong cross trait assortative mating between education and height in Scotland, and little in England. In this case, the SNP-phenotype associations in unrelated individuals would differ between Scotland and England, but the within family estimates would control for these differences and would reflect solely the individual level causal effects of the SNPs on each of the phenotypes. Therefore, all differences in population stratification, assortative mating, and dynastic effects should be controlled for in family designs. We have added the following to the discussion:

“Unlike analyses using summary data from unrelated individuals, two sample within family designs do not require the familial effects to be the same in the two samples. This is because the (different) familial effects in each sample are controlled for and the MR estimates use the individual level causal effect.”

2) The analytical models assume no confounding due to interaction between family structure and sibling-specific characteristics. For example, even after removing for family effects, residual confounding may still persist where this residual confounding is a product of family effects and individual-level characteristics. I think some light discussion on this matter could be useful.

The within family models cannot control for sibling-sibling interactions. If samples of parent-offspring quads are available, it should be possible to identify the effects of siblings on each other. However, this is not residual confounding (confounding of the transmission of variants from parents to offspring, i.e. confounding of the inheritance of germline DNA, is likely to be very rare). This is really a violation of the exclusion restriction, i.e. an alternative pathway through which the SNP affects the phenotype (via the sibling). Even these effects must be mediated via some phenotypic expression in the index sibling. However, the extent of these effects are unknown at present but we plan to investigate further in future studies as more data become available. We have noted this in the discussion:

“A limitation of sibling designs is that they must assume no sibling-sibling interaction effects.”

3) Is there a possibility that horizontal pleiotropy may reduce the effects due to confounding by family structure in your analytical models? I realize this concern is mostly theoretical, but I was curious if the authors have put some thought into this possibility.

Theoretically yes, e.g. if you had familial effects biasing estimates towards the null, and horizontal pleiotropy biasing the away from the null. However, we do not think there would be any way to detect this offsetting or use it systematically to know when an analysis in unrelated individuals was relatively unbiased. Therefore, while this is theoretically interesting, we think it would have limited use in empirical studies.